# WHAT HAPPENS WHEN GENERATIVE AI MODELS TRAIN RECURSIVELY ON EACH OTHER'S OUTPUTS?

**Hung Anh Vu, Galen Reeves, & Emily Wenger**[*]
Department of Electrical and Computer Engineering
Duke University

## ABSTRACT

The internet serves as a common source of training data for generative AI (genAI) models but is increasingly populated with AI-generated content. This duality raises the possibility that future genAI models may be trained on other models' generated outputs. Prior work has studied consequences of models training on their own generated outputs, but limited work has considered what happens if models ingest content produced by other models. Given society's increasing dependence on genAI tools, understanding such data-mediated model interactions is critical. This work provides empirical evidence for how data-mediated interactions might unfold in practice, develops a theoretical model for this interactive training process, and experimentally validates the theory. We find that data-mediated interactions can benefit models by exposing them to novel concepts perhaps missed in original training data, but also can homogenize their performance on shared tasks.

## 1 INTRODUCTION

Since the release of ChatGPT in 2022, generative AI (genAI) models have exploded in popularity. Now capable of generating highly realistic text, images, and videos, these models have been widely adopted for various use cases, from creative idea generation (Ali Elfa & Dawood, 2023) to healthcare support (Reddy, 2024) to national security settings (Harding, 2024; GAO, 2024). Given the significant uptick in genAI use across numerous industries, this technology is clearly here to stay. Consequently, interrogating potential ways genAI models could evolve—in positive or harmful ways—is critical.

With few exceptions, today's large-scale genAI models are trained on massive datasets sourced from the internet. Widely-accepted scaling laws for model performance say that training on more data aids learning (Kaplan et al., 2020), and the internet provides a rich, cheap, and ever-evolving source of training data. Although whitepapers for more recent genAI models withhold details about training set composition—potentially due to ongoing litigation about copyright concerns—evidence from earlier whitepapers indicates that scraped data was used to train models like Llama, Gemini, Phi, the GPT series, Claude, and others (Dubey et al., 2024; Achiam et al., 2023; Team et al., 2024a; Jiang et al., 2023; coh, 2024; Anthropic, 2023; Abdin et al., 2024).

Beyond privacy and copyright concerns, training on scraped data could have other downsides. Prior work has noted that genAI models trained recursively on their own generated outputs "collapse," becoming unable to generate meaningful content (Shumailov et al., 2024; Hataya et al., 2023; Martínez et al., 2023; Alemohammad et al., 2024). This scenario is feasible, since AI-generated content abounds online (Sun et al., 2025) and could be part of future training datasets. However, subsequent work has proposed ways to mitigate collapse via reuse of non-AI-generated data in subsequent training iterations (Dey & Donoho, 2024; Kazdan et al., 2025; Dohmatob et al., 2025; Feng et al., 2024). Model collapse remains an activate research area (Schaeffer et al., 2025).

Yet, prior work studying the dynamics of model collapse has overlooked another reality: the internet teems with content from *many* genAI models. Today's most popular models have millions of users (Reuters, 2024; Handa et al., 2025; AI, 2024a), who leverage generative AI tools to create online content like web pages and social media posts (gen, 2022). Recent work showed that up to 40% of content on popular sites like Quora is now AI-generated (Sun et al., 2025). Given the

---

[*]Corresponding author: `emily.wenger@duke.edu`

increasing availability of these models for a variety of public-facing uses, AI-generated content from many different models will continue to proliferate.

The standard practice of training on scraped internet data and the increasing prevalence of AI-generated content online suggest the strong possibility that *future generative AI models will be trained on other models' outputs*. Yet, this aspect of model training has received relatively little attention. Given the widespread adoption of generative AI models in critical settings like healthcare and national security, this phenomenon ought to be investigated to ensure models remain helpful and trustworthy.

**Contributions.** To address this need, this work theoretically derives and experimentally evaluates the long-term evolutionary behavior of generative models trained on *each other's* data. Specifically, we

- Develop a framework describing data-mediated interactions between genAI models.
- Derive concise formulas describing the dynamics of interactive training under varied regimes.
- Run experiments on large language models to understand how data-mediated interactions affect model performance in practice.

**Key findings.** Both our theoretical analysis and experiments show that when training with a mixture of real and synthetic data, the implicit interaction between heterogeneous models and datasets can have both positive and negative impacts. **Specifically, interactive recursive training can help models learn from each other's private data but risks homogenization.** At a high level, well known concepts in statistical learning theory anticipate this: recursive training on the same data is bad (e.g., overfitting and model "collapse") but training on novel data, even if synthetic, can boost performance (e.g., transfer learning). Our experimental results provide concrete evidence that these phenomena can occur simultaneously. Future work should further study these dynamics.

## 2 RELATED WORK

**Model collapse** is a recently observed phenomenon in large-scale generative text and image models. It referred—in its earliest form—to the phenomenon of models performing much worse after generations of training on their own generated outputs (Shumailov et al., 2024; Peterson, 2025; Wang et al., 2024; Alemohammad et al., 2024; Feng et al., 2024; Hataya et al., 2023; Dohmatob et al., 2025; Martínez et al., 2023). Theoretical and empirical results from these works show that models, if trained on generated outputs from their prior versions, slowly degrade in performance as generations progress. One way this manifests is in models forgetting the tails of their original (real) training data, since generated content tends not to contain rare content from the original training data. Training repeatedly on truncated, synthetic data leads the model to forget the richness of its original distribution, resulting in degraded performance (at best) and total failure (at worst).

**Mitigating model collapse.** Despite the dire predictions of these papers, subsequent work has proposed a simple mitigation strategy: instead of discarding all prior (human-generated) training data, retain some fraction of this while augmenting it with generated data. Numerous works have observed that this choice to *augment* instead of *discard* the original training dataset results in a bounded error in future models, avoiding collapse (Kazdan et al., 2025; Gerstgrasser et al., 2024; Marchi et al., 2024). Although most of these results were discovered on small models, recent work claims that the observed bound in error $\pi^2/6$ exists for all models (Dey & Donoho, 2024). Further work (Schaeffer et al., 2025) summarizes current research on collapse.

**Transfer learning and other model interactions.** Significant prior work has studied transfer learning, in which information learned by one model is passed to another, often by reusing the trained weights of a "teacher" model to initialize a "student" model (Zhuang et al., 2020). Some prior work has further considered the use of synthetic data in transfer learning (Tian & Shen, 2025; Kim et al., 2022; Brinner et al., 2025). Our work is distinct from transfer learning due to its focus on *unintentional* data-mediated interactions between models. Furthermore, limited work has examined long-term effects of models training on each other's data. Zhang et al. (2024) consider the setting where a generative model is trained on data generated by other models, but does not consider long-term effects of such interactions among multiple models. Jain & Krishnamurthy (2025) study interacting Large Language Model agents through the lens of Bayesian social learning and microeconomics, but do not focus specifically on data-mediated interactions between models.

Table 1: **Evidence from LLama, GPT, and Phi suggests reuse of old training and collection of additional data to train new model generations.** *Datasets reused across models and generations are highlighted. We start with Phi 1.5, the first version of Phi designed for general NLP tasks. Phi 1 was designed for coding tasks.*

| Model | v1 | v2 | v3 | v4 |
|---|---|---|---|---|
| Llama | (Touvron et al., 2023a): ArXiv, Books , Common Crawl , C4, Wikipedia, StackExchange | (Touvron et al., 2023b): "A new mix of publicly available online data." | (Dubey et al., 2024): "A variety of data sources containing knowledge until the end of 2023." | (AI, 2025): "A mix of publicly available, licensed data and information from Meta's products and services." |
| GPT | (Radford et al., 2018): BooksCorpus | (Radford et al., 2019): WebText | (Brown et al., 2020): CommonCrawl , WebText2 , Books, Books2 , Wikipedia | (Achiam et al., 2023): No info provided |
| Phi | (Li et al., 2023): The Stack , Stack Overflow , synthetic "textbook" data | (Abdin et al., 2023): The Stack , Stack Overflow , synthetic "textbook" data, filtered Commmon Crawl | (Abdin et al., 2024): "publicly available web data. . . and synthetic LLM-generated data" | N/A |

# 3 How Today's Large-Scale Generative AI Models Are Trained

We first establish *why* we believe that data-mediated interactions between models—e.g. instances of models training on each other's generated outputs—are realistic and worthy of study. To do this, we comb through academic literature and whitepapers describing today's large-scale genAI models to understand how models are trained, what data they are trained on, and how data is collected and used for model updates. This sets the stage for the formalization and experiments in the rest of the paper.

We find that most of today's models follow a 3 step update process. First, models are **pretrained** on a large corpus of data; then they are **fine-tuned** to teach specific behaviors and/or to align them with human preferences. Finally, they are later **updated**, either to teach new behaviors or update knowledge. As we describe these steps in detail below, we highlight specific realities or assumptions that have been largely overlooked or not made explicit by prior work.

**Step 1: Pretraining.** Following well-established scaling laws linking model performance and dataset size (Kaplan et al., 2020), large-scale generative AI models are trained on massive, internet-scraped datasets. Early versions of GPT, Llama, and PaLM all report being trained on scraped datasets like Common Crawl, ArXiv, Github, Wikipedia, and/or Stack Exchange (Touvron et al., 2023a; Chowdhery et al., 2023; Brown et al., 2020)—see Table 6 in Appendix for an overview.

Another striking fact emerges from the categorization of training data in Table 6: *large-scale model training datasets overlap.* For example, GPT, Jamba, Llama, PaLM, and Phi are all trained on subsets of CommonCrawl (Crawl, 2025), while GPT, Llama, and PaLM are all trained on Wikipedia and Books datasets. Several other models have other points of training data overlap.

**Step 2: Fine-tuning.** Variously called fine-tuning or alignment, *this phase leverages proprietary methods or data to tweak model behaviors in ways model providers believe are helpful.* For example, the LLama fine-tuning phase (Dubey et al., 2024) involves many rounds of reinforcement learning with human feedback (RLHF) to stamp out model negative behaviors, while Phi (Abdin et al., 2024) was fine-tuned on proprietary synthetic data to patch "gaps" in its mathematical reasoning abilities.

**Step 3: Model updates.** A key assumption of prior literature on model collapse is that models are *updated* by training on a mix of fresh and re-used data. The "replace" update scenario Shumailov et al. (2024) assumes model trainers train the next generation using only generated data outputted by the prior version of the model, an interesting but impractical setting Schaeffer et al. (2025). The "accumulate" update scenario (Gerstgrasser et al., 2024) assumes model trainers augment their original data at each update step with additional data that may contain AI-generated content. Finally, the "accumulate and subsample" update scenario (Kazdan et al., 2025) subsamples a fixed-size subset of original and accumulated data for each update, acknowledging real-world compute limits.

We believe that the "accumulate-and-subsample" update paradigm best reflects reality and so we leverage it in our work. We support this opinion with evidence from three well-documented model families: Llama, GPT, and Phi. Table 1 records the training data used in publicly disclosed generations of these models. As the table shows, *trainers re-use some prior training data for model updates, supplementing this with additional web content.* Whitepapers for models published after 2023 generally omit training data information but suggest collection of new online data for updates.

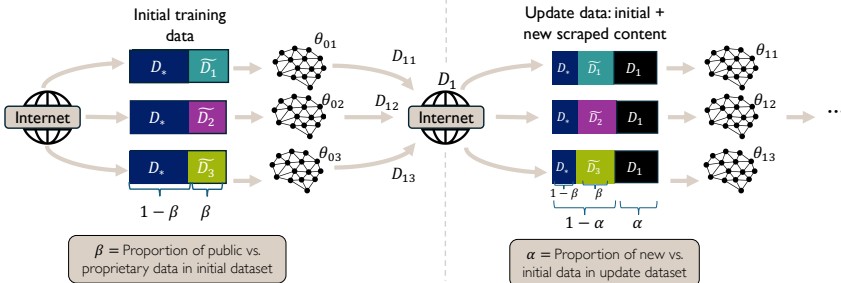

Figure 1: **Our dataset update scheme, parameterized by $\alpha$ and $\beta$.** *This paradigm best aligns with evidence from the literature given in §3 and strongly indicates that interactions between models, facilitated by training on others' generated data, are an important consideration for empirical and theoretical work on model evolution.*

Despite these realities, most prior work on model collapse still overlooks a fundamental reality in model updates: *future models trained on internet-sourced content will be trained on outputs from other generative AI models*, not merely their own. Already, the internet is filled with generated content from various models (Sun et al., 2025). As model trainers collect new data to facilitate model updates, internet-sourced data will inevitably contain content from other generative models, so:

> At each update step, models may be trained on *their own* and on *other models'* generated outputs.

## 4  FORMALIZING AN ITERATIVE, INTERACTIVE MODEL TRAINING PIPELINE

Section 3 provides empirical evidence for two realities overlooked by prior studies: internet-scraped training datasets used for initial training may have substantial overlap, and models may be updated using *each others' generated outputs*. To study the effect of these two factors on model evolution, we propose a general workflow in which multiple entities regularly update their models using a mix of private, public, and generated data. Based on §3, we consider three types of training/update data:

- $D_*$: Public data used during initial training/updates by multiple entities (real data only).
- $\tilde{D}_k$: Private data used only by entity $k$ for initial training/updates (real data only).
- $D_t = \{D_{t1}, D_{t2}, \ldots D_{tk}\}$: Public data used for updates at time $t$ by multiple entities (synthetic data). $D_{tk}$ is data generated by the $k^{th}$ entity based on model $\theta_{t-1,k}$.

Mapping these to realistic scenarios, $D_*$ could be a public dataset like Common Crawl; $\tilde{D}_k$ could be a private dataset of math problems curated by entity $k$; and $D_t$ could be an internet scrape from after initial model training. We weight the relative impact of these data types by the ratios $\alpha, \beta$.

- $\beta$, $0 \leq \beta \leq 1$, is relative size of the initial public data set $D_*$ compared to the initial private data set $\tilde{D}_k$. This fraction remains constant if/when initial data is reused for updates.
- $\alpha$, $0 \leq \alpha \leq 1$ is the fraction of new data introduced at generation $t$, relative to the amount of initial data reused (following the "accumulate and subsample" paradigm of Kazdan et al. (2025)).

**Interactive training workflow.** We consider $K$ entities, each seeking to train or update its own generative AI model. In the initial phase of training, denoted by time $t = 0$, each entity $k$ trains its model based on a combination of a publicly available dataset $D_*$ as well its own private dataset $\tilde{D}_k$. The trained model is represented generally by a parameter $\hat{\theta}_{t,k}$, i.e., $\hat{\theta}_{0,k} = \Phi_{0,k}(D_*, \tilde{D}_k)$, $k = 1, \ldots, K$. where each $\Phi_{0,k}$ represents a generic training algorithm. For model updates at stages $t > 0$, model parameters are updated via:

1. New public data $D_t$ is generated uniformly at random using the most recent version of the models. Specifically, the data are sampled i.i.d. according to the mixture $\frac{1}{K} \sum_{k=1}^{K} P_{k,\hat{\theta}_{t-1,k}}$ where $P_{k,\theta}$ denotes the generative model used by $k$-th entity.
2. This data is placed online and collected by entities as training data for the next model update.
3. Each entity composes its training data for the next update, using a mix of the initial dataset $(D_*, \tilde{D}_k)$ and newly collected data $D_t$. Contributions from each dataset are weighted by $\alpha, \beta$.

4. Each entity $k = 1, \ldots, K$ updates it model parameters via $\hat{\theta}_{t,k} = \Phi_{t,k}(\hat{\theta}_{t-1,k}, D_*, \tilde{D}_k, D_t)$. Here, $\Phi_{0,k}$ is a training algorithm that depends on the previous model parameter $\hat{\theta}_{k,t-1}$ as well as the data. As before, training may employ subsampling, weighting, and randomization.

In this workflow, entities interact through the release of publicly available synthetic data produced by prior generations of other entities' models. Thus, even though initial private training data are never shared, it could end up positively impacting other entities' models. This potential benefit of synthetic data sharing appears only in this interaction paradigm and has not been recognized in prior work.

## 5 THEORY

We theoretically analyze the behavior of the interactive workflow. Similar to prior work (Gerstgrasser et al., 2024; Kazdan et al., 2025; Dey & Donoho, 2024; Dohmatob et al., 2025; Barzilai & Shamir, 2025), we focus on the linear regression models where each data point consists of a feature-response pair $(x, y) \in \mathbb{R}^d \times \mathbb{R}$. By the universality results of Dey & Donoho (2024), the analysis of this settings also applies to generalized linear models satisfying appropriate asymptotic normality assumptions.

**Notation.** For a $p \times q$ matrix $A$, we use $A^+$ to denote the Moore-Penrose pseudoinverse and $\text{vec}(A)$ to denote the $pq \times 1$ vector obtained by stacking the columns. $\otimes$ denotes the Kronecker product. For $\alpha, \beta \in [0, 1]$ we set $\bar{\alpha} = 1 - \alpha$ and $\bar{\beta} = 1 - \beta$.

**Training Workflow.** We follow the training pipeline outlined in Section 4 in which $K$ different models are trained on a mixture of private, public, and generated data. At initialization, each entity $k \in [K]$ combines its private data $\tilde{D}_k = (\tilde{x}_{ki}, \tilde{y}_{ki})_{i=1}^{\tilde{n}_k}$ with public data $D_* = (x_{*i}, y_{*i})_{i=1}^{n_*}$ to produce an estimate $\hat{\theta}_{k0}$ by minimizing the empirical loss

$$\sum_{(x,y) \in \tilde{D}_k} \beta_0 L(x, y, \theta) + \sum_{(x,y) \in \tilde{D}_*} \bar{\beta}_0 L(x, y, \theta)$$

where $L(x, y, \theta) := (y - x^\top \theta)^2$ is the squared error loss and $0 \leq \beta_0 \leq 1$ controls the relative weight placed on the private data. Training then proceeds for generation stages $t = 1, 2, 3, \ldots$ as follows:

1. Each entity $k$ uses its most recent parameter estimate $\hat{\theta}_{t-1,k}$ to generate new data $D_{tk} = (x_{tki}, y_{tki})_{i=1}^{n_{tk}}$ according to the Gaussian model $y \mid x \sim \mathsf{N}(x^\top \theta_{t-1,k}, \sigma^2)$. The entire collection of generated samples is combined into a single public data set $D_t = \cup_{k=1}^{K} D_{tk}$.

2. Each entity $k$ produces a new estimate $\hat{\theta}_{tk}$ by minimizing the empirical loss

$$\sum_{(x,y) \in D_k} \bar{\alpha}_t \beta_t L(x, y, \theta) + \sum_{(x,y) \in D_*} \bar{\alpha}_t \bar{\beta}_t L(x, y, \theta) + \sum_{(x,y) \in D_t} \frac{\alpha_t}{K} L(x, y, \theta)$$

with weights $0 \leq \alpha_t, \beta_t \leq 1$.

We note that our framework could easily be extended to accommodate new, human-generated data at each time step, but we omit this in our formulation for analytic simplicity. Throughout our analysis we assume that all features are deterministic. We represent dataset $\tilde{D}_k$ with $\tilde{n}_k \times d$ matrix $\tilde{X}_k = [\tilde{x}_{k1}, \ldots, \tilde{x}_{k\tilde{n}_k}]^\top$ and $\tilde{n}_k \times 1$ vector $\tilde{y}_k = [\tilde{y}_{k1}, \ldots, y_{k\tilde{n}_k}]^\top$, and use the same convention for the public data $(X_*, y_*)$ and the generated data $(X_{tk}, y_{tk})$. Data across different entities are then combined into "lifted" representations, which are denoted using boldface:

$$\tilde{\boldsymbol{X}} = \begin{bmatrix} \tilde{X}_1 & & \\ & \ddots & \\ & & \tilde{X}_K \end{bmatrix}, \quad \boldsymbol{y}_0 = \begin{bmatrix} \tilde{y}_1 \\ \vdots \\ \tilde{y}_K \end{bmatrix}, \quad \boldsymbol{X}_t = \begin{bmatrix} X_{t1} & & \\ & \ddots & \\ & & X_{tK} \end{bmatrix}, \quad \boldsymbol{y}_t = \begin{bmatrix} y_{t1} \\ \vdots \\ y_{tK} \end{bmatrix}$$

We note that information about which entity produced which sample is required for the analysis, but is not used during the training, where all data from the same generation are treated interchangeably.

**Bias-Variance Decomposition.** We derive exact formulas for the mean and variance of the estimators at each stage of the workflow. Given the features $(\tilde{\boldsymbol{X}}, X_*, \boldsymbol{X}_t)$ and learning weights $(\alpha_t, \beta_t)$ define

$$\tilde{\boldsymbol{S}} = \text{diag}(\tilde{S}_1, \ldots, \tilde{S}_k) := \tilde{\boldsymbol{X}}^\top \tilde{\boldsymbol{X}}, \qquad S_* := X_*^\top X_* \qquad \boldsymbol{S}_t = \text{diag}(S_{t1}, \ldots, S_{tk}) := \boldsymbol{X}_t^\top \boldsymbol{X}_t$$

$$\boldsymbol{G}_t := \bar{\alpha}_t \beta_t \tilde{\boldsymbol{S}} + \bar{\alpha}_t \bar{\beta}_t (\mathrm{I}_K \otimes S_*) + \alpha_t (\mathrm{I}_K \otimes \underline{\boldsymbol{S}}_t) \qquad \underline{\boldsymbol{S}}_t := \frac{1}{K} \sum_{k=1}^{K} S_{tk}$$

$$\boldsymbol{P}_t := \bar{\alpha}_t \boldsymbol{G}_t^+ \begin{bmatrix} \beta_t \tilde{\boldsymbol{S}} & \bar{\beta}_t (\mathbf{1}_K \otimes S_*) \end{bmatrix} \qquad \boldsymbol{Q}_t := \alpha_t \boldsymbol{G}_t^+ \Pi \boldsymbol{S}_t$$

where $\Pi := \frac{1}{K}(\mathbf{1}_{K \times K} \otimes \mathrm{I}_d)$ is an orthogonal projection matrix and $\alpha_0 \equiv 0$.

---

**Theorem 1.** *Conditional on the initial data $D_0 := (\tilde{D}_1, \ldots, \tilde{D}_K, D_*)$, the estimates $\hat{\boldsymbol{\theta}}_t = \mathsf{vec}(\hat{\theta}_{t1}, \ldots, \hat{\theta}_{tK})$ are Gaussian with mean and variance*

$$\mathbb{E}[\hat{\boldsymbol{\theta}}_t \mid D_0] = \boldsymbol{M}_t \begin{bmatrix} \tilde{\boldsymbol{X}}^+ \tilde{\boldsymbol{y}} \\ X_*^+ y_* \end{bmatrix}, \qquad \mathsf{Cov}(\hat{\boldsymbol{\theta}}_t \mid D_0) = \boldsymbol{C}_t$$

*where the matrices $\boldsymbol{M}_t$ and $\boldsymbol{C}_t$ are defined recursively with $\boldsymbol{M}_0 = \boldsymbol{P}_0$ and $\boldsymbol{C}_0 = \boldsymbol{0}_{Kd \times Kd}$ and*

$$\boldsymbol{M}_t = \boldsymbol{P}_t + \boldsymbol{Q}_t \boldsymbol{M}_{t-1}, \qquad \boldsymbol{C}_t = \boldsymbol{Q}_t(\sigma^2 \boldsymbol{S}_t^+ + \boldsymbol{C}_{t-1})\boldsymbol{Q}_t, \qquad t \geq 1.$$

---

Theorem 1 shows that the conditional mean of each estimate $\hat{\theta}_{tk}$ is a linear combination of the individual ordinary least squares (OLS) estimates $\tilde{X}_1^+ \tilde{y}_1, \ldots, \tilde{X}_K^+ \tilde{y}_K$ and $X_*^+ y_*$ for the private data and public data, respectively. For each generation $t$, similarity across entities can be assessed by comparing the rows of the $K \times (K+1)$ block partitioning of $\boldsymbol{M}_t$. At initialization, the off-diagonal blocks for the private data are zeroed out, but in later stages, these blocks become nonzero thereby allowing private data to be shared across entities. Homogenization (i.e., shrinkage towards a global consensus) occurs when row blocks are identical, and thus each entity has the same mean.

For our next result we mimic the experimental setup in Section 6 and assume that the initial data are generated from a Gaussian model with a common ground truth parameter and the heterogeneity across datasets arises from the differences in the features, i.e., the matrices $\tilde{S}_1, \ldots, \tilde{S}_K$.

---

**Theorem 2.** *Suppose that the initial data are generated independently according to the model $y \mid x \sim \mathsf{N}(x^\top \theta, \sigma^2 \mathrm{I}_d)$ where $\theta \in \mathbb{R}^d$ is a fixed parameter. If $\boldsymbol{G}_1, \ldots, \boldsymbol{G}_t$ are full rank then*

$$\mathbb{E}[\hat{\boldsymbol{\theta}}_t] = \big(\mathrm{I} - \boldsymbol{Q}_t \cdots \boldsymbol{Q}_1(\mathrm{I} - \boldsymbol{G}_0 \boldsymbol{G}_0^+)\big)(\mathbf{1}_K \otimes \theta), \quad \mathsf{Cov}(\hat{\boldsymbol{\theta}}_t) = \boldsymbol{M}_t \begin{bmatrix} \tilde{\boldsymbol{S}}^+ & 0 \\ 0 & S_*^+ \end{bmatrix} \boldsymbol{M}_t^\top + \boldsymbol{C}_t$$

---

To help interpret this result, observe that if $\boldsymbol{G}_0$ is full rank, then each initial estimate is unbiased, and unbiasedness persists throughout every stage of training. Conversely, if $\boldsymbol{G}_0$ is rank deficient, then at least one (and possibly all) of the initial estimates is biased. Remarkably, Theorem 2 shows that it may still be possible for all entities have vanishing bias, provided that $\boldsymbol{Q}_t \cdots \boldsymbol{Q}_s$ converges to zero. Specific conditions under which this occurs are considered in the next section.

**Asymptotic Variance.** To provide a finer analysis of the training dynamics we now suppose that the weights and features satisfy $\alpha_t = \alpha$, $\beta_t = \beta$, and $\boldsymbol{S}_t = \boldsymbol{S}$ for $t \geq 1$. Setting $\boldsymbol{P} = \boldsymbol{P}_1$ and $\boldsymbol{Q} = \boldsymbol{Q}_1$, the matrices $\boldsymbol{M}_t$ and $\boldsymbol{C}_t$ defined in Theorem 1 can be expressed explicitly as

$$\boldsymbol{M}_t = \boldsymbol{Q}^t \boldsymbol{P}_0 + \Big(\sum_{s=0}^{t-1} \boldsymbol{Q}^s\Big)\boldsymbol{P}, \qquad \boldsymbol{C}_t = \sigma^2 \sum_{s=1}^{t} \boldsymbol{Q}^s \boldsymbol{S}^+ (\boldsymbol{Q}^s)^\top \tag{1}$$

Classical results in matrix analysis (Higham, 2002) imply that if the spectral radius of $\boldsymbol{Q}$ is strictly less than one, then these matrices converge to well-defined limits $\boldsymbol{M}$ and $\boldsymbol{C}$ satisfying

$$\boldsymbol{M} := (\mathrm{I} - \boldsymbol{Q})^{-1}\boldsymbol{P}, \qquad \mathsf{vec}(\boldsymbol{C}) := \sigma^2(\mathrm{I} - \boldsymbol{Q} \otimes \boldsymbol{Q})^{-1}\mathsf{vec}(\boldsymbol{Q}\boldsymbol{S}^+\boldsymbol{Q}) \tag{2}$$

The following result provides a sufficient condition for convergence in terms of the triple $(\tilde{\boldsymbol{S}}, S_*, \boldsymbol{S})$. In particular, if $\boldsymbol{S}$ is proportional to $\tilde{\boldsymbol{S}}$ then the condition is satisfied for all $0 \leq \alpha < 1$ and $0 \leq \beta \leq 1$. Note that the boundary case $\alpha = 1$ corresponds to the recursive training setting of Shumailov et al. (2024) where the variance increases linearly across generations, and thus convergence does not occur.

**Lemma 1.** *Suppose that $\boldsymbol{S} \propto \lambda \tilde{\boldsymbol{S}} + (1 - \lambda)(\mathrm{I}_K \otimes S_*)$ for some $0 < \lambda \leq 1$. Then, the spectral radius of $\boldsymbol{Q}$ is strictly less than one for all $0 \leq \alpha < 1$ and $0 < \beta \leq \lambda$.*

We summarize our findings with the following characterization of the asymptotic variance:

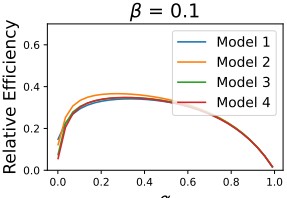 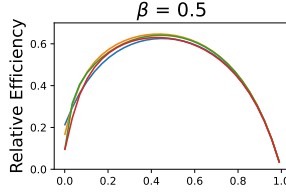 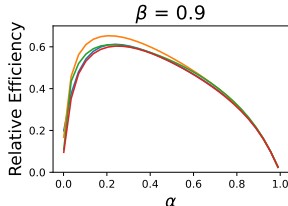

Figure 2: **Predicted relative efficiency across** $\alpha, \beta$ **values for a** $K = 4$ **model system (dimension** 15 **and rank** 5**).** *Curves show ratio of MSE of the minimum variance unbiased estimator to asymptotic MSE for a given* $\alpha, \beta$ *setting obtained from Theorem 3 (1 is optimal). Setting* $\alpha = \beta = 0.5$ *produces best results across models.*

---

**Theorem 3.** *Consider the setting of Theorem 2 and suppose that* $\alpha_t = \alpha$, $\beta_t = \beta$, *and* $\boldsymbol{S}_t = \boldsymbol{S}$ *for* $t \geq 1$. *If* $\boldsymbol{G} = \boldsymbol{G}_1$ *has full rank and* $\boldsymbol{Q}$ *has spectral radius strictly less than one, then*

$$\mathbb{E}[\boldsymbol{\theta}_t] \xrightarrow{t \to \infty} \mathbf{1}_K \otimes \theta, \qquad \mathsf{Cov}(\hat{\boldsymbol{\theta}}_t) \xrightarrow{t \to \infty} \sigma^2 \boldsymbol{M} \begin{bmatrix} \tilde{\boldsymbol{S}}^+ & 0 \\ 0 & S_*^+ \end{bmatrix} \boldsymbol{M}^\top + \boldsymbol{C}$$

*where* $\boldsymbol{M}$ *and* $\boldsymbol{C}$ *are given by* (2).

---

**MSE and relative efficiency.** The expression for the mean and variance in Theorems 2 and 3 provide explicit formulas for the mean squared error (MSE) $\mathbb{E}[\|\hat{\theta}_{tk} - \theta\|^2]$ of entity $k$ at each generation $t$ and the mean squared prediction error (MSPE) $\mathbb{E}[\|\tilde{X}_m(\hat{\theta}_{tk} - \theta)\|^2]$ for entity $m$'s private feature matrix.

We can use this to compute the optimal $\alpha, \beta$ values for the interactive training setting. Figure 2 compares the asymptotic MSE for a training workflow with given $\alpha, \beta$ (from Theorem 3) with the MSE of an idealized setting where each entity has access to the entire collection of real data (both private and public). Each curve represents the relative efficiency, i.e., the ratio of optimal MSE to entity-specific workflow MSE, with values close to one indicating near optimality. These results demonstrate that a setting with $\beta = 0.5$ and $\alpha = 0.5$ achieves the best global performance for all models. When $\beta$ is much larger (0.9), relatively small $\alpha$ values also improve model performance.

## 6 EXPERIMENTAL EVALUATION

To understand how our theoretical predictions bear out in practice, we run experiments on text-generation models. In each, we train $K$ interacting models (per our framework in Figure 1) for several generations and evaluate how their performance changes on their own and other models' tasks. Here, we present the setup and results for a $K = 2$ and $K = 3$ interacting at $\beta = 0.5$ model system. See Appendix for full $K = 2$ results at various beta values.

Table 2: **Change in loss behavior for** $K = 2$ **interacting models at** $\beta = 0.5$. *We show results as* `initial` $\to$ `final` *prediction loss values for models on their own and the other models' tasks on their own and the other models' tasks over* $T$ *generations. For clarity, we colorize loss* increase *,* decrease *, and* constancy *(*$\Delta \leq 0.1$*).*

(a) *OPT models (T=15)*

| | $\alpha = 0$ | $\alpha = 0.5$ | $\alpha = 1.0$ |
|---|---|---|---|
| Model 1 on Task 1 | $3.3 \to 3.3$ | $3.3 \to 3.3$ | $3.3 \to 3.5$ |
| Model 2 on Task 2 | $1.8 \to 1.7$ | $1.8 \to 1.7$ | $1.8 \to 2.2$ |
| Model 1 on Task 2 | $3.1 \to 3.5$ | $3.1 \to 1.8$ | $3.1 \to 2.2$ |
| Model 2 on Task 1 | $5.1 \to 5.1$ | $5.1 \to 3.5$ | $5.1 \to 3.5$ |

(b) *Llama 3.2 (1B) models (T=15)*

| | $\alpha = 0$ | $\alpha = 0.5$ | $\alpha = 1.0$ |
|---|---|---|---|
| Model 1 on Task 1 | $2.8 \to 2.9$ | $2.8 \to 2.9$ | $2.8 \to 3.0$ |
| Model 2 on Task 2 | $1.2 \to 1.2$ | $1.2 \to 1.3$ | $1.2 \to 1.9$ |
| Model 1 on Task 2 | $2.0 \to 2.5$ | $2.0 \to 1.4$ | $2.0 \to 1.8$ |
| Model 2 on Task 1 | $3.7 \to 4.2$ | $3.7 \to 3.0$ | $3.7 \to 3.0$ |

(c) *Llama 3.2 (3B) models (T=8)*

| | $\alpha = 0$ | $\alpha = 0.5$ | $\alpha = 1.0$ |
|---|---|---|---|
| Model 1 on Task 1 | $2.4 \to 2.5$ | $2.4 \to 2.5$ | $2.4 \to 2.6$ |
| Model 2 on Task 2 | $1.0 \to 1.0$ | $1.0 \to 1.0$ | $1.0 \to 1.3$ |
| Model 1 on Task 2 | $1.8 \to 2.1$ | $1.8 \to 1.2$ | $1.8 \to 1.3$ |
| Model 2 on Task 1 | $3.5 \to 3.8$ | $3.5 \to 2.6$ | $3.5 \to 2.7$ |

**Experiment setup.** Training large language models from scratch is computationally infeasible for us, so we simulate the initial setup of language models trained on dataset $(D_*, \tilde{D}_k)$ at time $t = 0$ by fine-tuning $K = 2$ instances of a given pre-trained model architecture on carefully chosen $D_*, \tilde{D}_k$. We experiment with two language model architectures—OPT-350m (Zhang et al., 2022), Llama 3.2 1B (Dubey et al., 2024), and Llama 3.2 3B (Dubey et al., 2024)—to assess how interactive training scales. Public sources (Touvron et al., 2023a; Zhang et al., 2022) state that both models were pretrained on BookCorpus (Zhu et al., 2015), CC-Stories (Anderson, 2022), the English portion of CommonCrawl, and public Reddit data. We approximate $D_*$ with BookCorpus due to practical constraints. Each model is given its own initial task-specific dataset $\tilde{D}_k$ —SciQ (Johannes Welbl,

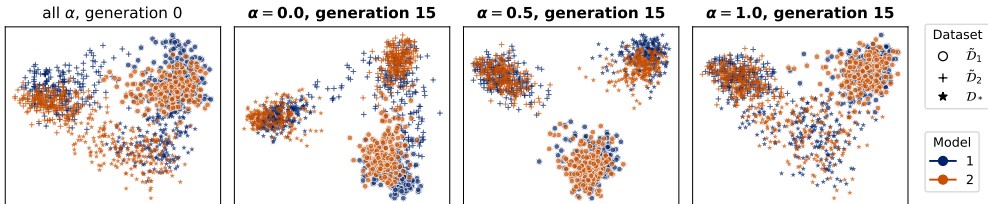

Figure 3: **PCA of embeddings of outputs produced by models** $\theta_{t1}$ **and** $\theta_{t2}$ **on datasets** $\tilde{\mathcal{D}}_1$, $\tilde{\mathcal{D}}_2$, **and** $\tilde{\mathcal{D}}_*$. **Results for** $K = 2$ **Llama models trained with** $\beta = 0.5$ **and varying** $\alpha$. *Leftmost plot shows embeddings at generation $t = 0$, which are identical for all $\alpha$; while right plots show embeddings at $t = 15$ with different $\alpha$.*

Table 3: **Cosine similarity of embedded output representations for** $K = 2$ **Llama models at** $\beta = 0.5$. *We show* `initial(t=0)` $\to$ `final(t=15)` *for* $\tilde{\mathcal{D}}_1$ $\tilde{\mathcal{D}}_2$, *and* $\mathcal{D}_*$. *We colorize* increase *and* decrease .

|  | $\alpha = 0$ | $\alpha = 0.5$ | $\alpha = 1.0$ |
|---|---|---|---|
| $\tilde{\mathcal{D}}_1$ | $0.73 \to 0.70$ | $0.73 \to 0.88$ | $0.73 \to 0.86$ |
| $\tilde{\mathcal{D}}_2$ | $0.88 \to 0.79$ | $0.88 \to 0.93$ | $0.88 \to 0.94$ |
| $\mathcal{D}_*$ | $0.50 \to 0.59$ | $0.50 \to 0.64$ | $0.50 \to 0.58$ |

2017) (science questions) for the $k = 0$ model; and OpenAI's GSM8K (Cobbe et al., 2021) (grade school level math problems) for the $k = 1$ model. When simulating $K = 3$, we assign the third model the AI2 ARC dataset (reasoning questions) (Clark et al., 2018).

Interactive training proceeds as outlined in Figure 1, with fixed $\alpha$, $\beta$ values for each experiment. After training a new model generation $\hat{\theta}_{tk}$, we use $\hat{\theta}_{tk}$ to produce synthetic data $D_{t+1,k}$ that becomes part of the next generation's training data (if $\alpha > 0$). We produce $D_{t+1,k}$ by randomly sampling prompts from $\tilde{D}_k$ for each of the $K$ models and prompting $\hat{\theta}_{tk}$ to complete the text.

**Training and evaluation.** We run experiments on $K = 2$ model systems with $\alpha \in \{0, 0.5, 1\}$ and $\beta \in \{0, 0.5, 1\}$, each for $T = 15$ generations of training. At each training generation, models are fine-tuned on datasets of fixed size $n = 12,500$ drawn i.i.d. from the datasets $\tilde{D}_k$, $D_*$, $D_t$ with weights $\bar{\alpha}\beta$, $\bar{\alpha}\bar{\beta}$, and $\alpha/K$, respectively. This mimics the *accumulate and subsample* setup of Kazdan et al. (2025) with the additional wrinkle of data-mediated model interactions.

We train each model for 100 steps per generation on a single NVIDIA H200 GPU using mixed-precision, the AdamW optimizer with a learning rate of $8e^{-6}$, warmup ratio of 0.025, and gradient accumulation over 2 steps. A table with training hyperparameters is in the Appendix. After training each generation, we record token-wise average cross-entropy loss by feeding each model prompts from each test set of private data $\tilde{D}_k$ and evaluating semantic "distance" between predicted and correct answer. We also compute embedded representations of models' completions of the first 200 elements of each of $\tilde{D}_k$ and $D_*$, to see how various $\alpha$, $\beta$ affect models' representational spaces. Embeddings are computed via the `SentenceTransformers` python library. If models produce outputs with similar embeddings (measured via cosine similarity), their feature spaces are more aligned. These two metrics allow us to evaluate how data-mediated interactions affect (1) models' performance on their own and other models' tasks and (2) model homogeneity.

**Results.** As predicted in Figure 2, an $\alpha = 0.5$, $\beta = 0.5$ setting produces optimal results in terms of model performance across tasks. Table 2 reports the change in models' loss values between the first and last training generation for the $\beta = 0.5$ setting. When $\alpha = 0$, the same (human-generated) data is used for each training update, resulting in stable or slightly worse performance on different tasks, as models are forced into a local minimum. When $\alpha = 1.0$, models are only trained on generated outputs. They degrade on their original task due to the lack of real data but improve slightly on the other model's task. However, at $\alpha = 0.5$, models perform well on their own tasks and improve on the other model's task. Table 4 shows similar results when we scale up $K = 2 \to K = 3$, introducing more diversity. Results for other $\beta$ and $K = 3$ are in Appendix and echo findings here.

While some amount of mixing improves model performance on previously-unseen tasks, homogenization occurs for $D_*$ at all $\alpha$ and for $\tilde{D}_k$ when $\alpha > 0$. These are the settings under which models share information, either via common dataset $D_*$ or interactive training when $\alpha > 0$. Figure 3 shows

the benefit of interactive training when $\alpha, \beta$ are reasonable, visualizing PCA-reduced embeddings from $\tilde{D}_1, \tilde{D}_2$, and $D_*$ for $\beta = 0.5$ and varying $\alpha$ as training progresses For $\alpha = 0.5$, the PCA shows clearly separated task clusters, indicating that both models are better tailored to the individual tasks. In contrast, the lack of well-defined clusters when $\alpha \neq 0.5$ suggests reduced sensitivity to task type, e.g. generic answers. Yet, in Table 3, we see that when $\alpha = 0$, models homogenize slightly on the shared task $D_*$ but diverge on model-specific tasks. This makes sense since models the models do not train on each other's tasks. Once $\alpha > 0$, homogenization increases for all datasets/tasks.

Table 4: **Change in loss behavior for $K = 3$ interacting models at $\beta = 0.5$.** *We show results as* `initial →` `final` *prediction loss values for models on their own and the other models' tasks over $T$ generations. For clarity, we colorize loss* increase *,* decrease *, and* constancy *($\Delta \leq 0.1$).*

(a) *OPT models (T=15)*

| | $\alpha = 0$ | $\alpha = 0.5$ | $\alpha = 1.0$ |
|---|---|---|---|
| Model 1 on Task 1 | $3.3 \to 3.3$ | $3.3 \to 3.3$ | $3.3 \to 3.5$ |
| Model 2 on Task 2 | $1.8 \to 1.7$ | $1.8 \to 1.7$ | $1.8 \to 2.3$ |
| Model 3 on Task 3 | $2.8 \to 2.9$ | $2.8 \to 2.9$ | $2.8 \to 3.1$ |
| Model 1 on Task 2 | $3.1 \to 3.5$ | $3.1 \to 1.9$ | $3.1 \to 2.2$ |
| Model 2 on Task 3 | $3.9 \to 4.2$ | $3.9 \to 3.1$ | $3.9 \to 3.1$ |
| Model 3 on Task 1 | $3.7 \to 3.8$ | $3.7 \to 3.4$ | $3.7 \to 3.6$ |
| Model 1 on Task 3 | $3.3 \to 3.6$ | $3.3 \to 3.0$ | $3.3 \to 3.1$ |
| Model 2 on Task 1 | $5.1 \to 5.1$ | $5.1 \to 3.5$ | $5.1 \to 3.5$ |
| Model 3 on Task 2 | $2.9 \to 2.9$ | $2.9 \to 1.9$ | $2.9 \to 2.2$ |

(b) *Llama 3.2 (1B) models (T=8)*

| | $\alpha = 0$ | $\alpha = 0.5$ | $\alpha = 1.0$ |
|---|---|---|---|
| Model 1 on Task 1 | $2.6 \to 2.7$ | $2.6 \to 2.7$ | $2.6 \to 2.8$ |
| Model 2 on Task 2 | $1.1 \to 1.1$ | $1.1 \to 1.2$ | $1.1 \to 1.4$ |
| Model 3 on Task 3 | $2.3 \to 2.8$ | $2.3 \to 2.6$ | $2.3 \to 2.5$ |
| Model 1 on Task 2 | $2.0 \to 2.4$ | $2.0 \to 1.3$ | $2.0 \to 1.4$ |
| Model 2 on Task 3 | $3.1 \to 3.4$ | $3.1 \to 2.5$ | $3.1 \to 2.5$ |
| Model 3 on Task 1 | $3.1 \to 3.7$ | $3.1 \to 3.0$ | $3.1 \to 2.8$ |
| Model 1 on Task 3 | $2.9 \to 3.1$ | $2.9 \to 2.5$ | $2.9 \to 2.5$ |
| Model 2 on Task 1 | $3.8 \to 4.2$ | $3.8 \to 2.8$ | $3.8 \to 2.8$ |
| Model 3 on Task 2 | $2.0 \to 2.5$ | $2.0 \to 1.3$ | $2.0 \to 1.4$ |

## 7 DISCUSSION

**Limitations.** Our work has several limitations. First, our theory considers only linear models, which, while common in the model collapse literature, may not capture nuances present in larger models. Second, we run experiments on LLMs in controlled settings, the dynamics of which may differ from real-world LLMs. Third, arguments against the increasing presence of generated outputs in training datasets (e.g. (Drayson et al., 2025)) are in Appendix B. Also, our theoretical framework assumes that new data in model updates is purely synthetic. In reality, if internet scrapes are used to create model update datasets, they will contain both synthetic and real data. Finally, we assume that model trainers use new scraped data for each model update but only reuse data from initial training. This assumption may limit the range of outcomes. Although our framework can accommodate an arbitrary number of interacting entities, exploring configurations with large K and T was computationally infeasible, as each setting requires repeated fine-tuning and evaluation of multiple large models. Therefore, we restricted our experiments to smaller practically achievable scales that preserve interpretability.

**Broader Impacts.** If data-mediated interactions homogenize generative models, causing them to coalesce on certain viewpoints, this could lead to pervasive bias in AI-generated content. Peterson (2025) discusses this possibility, while Wenger & Kenett (2025) showed homogeneity across creative outputs from many LLMs, suggesting these homogenization effects may already be felt. Much future study is needed to evaluate the extent to which data-mediated interactions fuel homogeneity.

**Conclusions and Future Work.** We provide a first look at possible outcomes of genAI models trained on each others' data and find mixed effects. Training on other models' data exposes models to concepts possibly missed in their own training data, but can homogenize model behaviors. Future work could consider additional nuances of interactions between models, explore how these interactions evolve in other modalities like image generation, and investigate whether fixed points (e.g. like the universal $\pi^2/6$ pathway of (Dey & Donoho, 2024)) exist under this paradigm.

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

# Appendix

## A    EXACT TRAINING DATA STATEMENTS FROM LLM PAPERS

Table 5 lists statements made about model training and fine-tuning data for large-scale generative AI models that do not explicitly list training data sources.

| Model | Pre-training data | Fine-tuning data |
|---|---|---|
| Claude 2 (Anthropic, 2023) | Claude models are trained on a proprietary mix of publicly available information from the Internet, datasets that we license from third party businesses, and data that our users affirmatively share or that crowd workers provide. | Publicly released on HuggingFace (Anthropic). |
| GPT4+ (Achiam et al., 2023) | No information provided | No information provided |
| Grok 3 (xAI, 2025) | No information provided | No information provided |
| Jamba (Team et al., 2024b) | Our pre-training dataset is a mixture of publicly available web documents, code, books and scientific articles. | When performing supervised fine-tuning, we make heavy use of synthetic data. |
| Llama 3 (Dubey et al., 2024) | We create our dataset for language model pre-training from a variety of data sources containing knowledge until the end of 2023. Much of the data we utilize is obtained from the web | We produce the aligned Llama 3 models by applying several rounds of post-training, or aligning the model with human feedback. |
| Llama 4 (AI, 2025) | A mix of publicly available, licensed data and information from Meta's products and services. This includes publicly shared posts from Instagram and Facebook and people's interactions with Meta AI. | No information provided. |
| Phi 3 (Abdin et al., 2024) | Our training data of consists of heavily filtered publicly available web data . . . from various open internet sources, as well as synthetic LLM-generated data. | [Supervised fine tuning] leverages highly curated high-quality data across diverse domains, e.g., math, coding, reasoning, conversation, model identity, and safety. |

Table 5: **Exact wording of training and fine-tuning data discussion from whitepapers in which data sources are not explicitly listed.**

Table 6: **Examples of training data listed for prominent language models.** ●= *use explicitly stated;* ◐= *significant overlap expected (e.g. GLaM (Du et al., 2022) and PaLM (Chowdhery et al., 2023) are trained on Microsoft's internal re-creation of WebText (Brown et al., 2020)). We only include models for which training data sources are explicitly stated. See Table 5 in Appendix for information on other prominent models.*

| Model | CommonCrawl | WebText | Github | Wikipedia | Books | ArXiv | StackExchange | News |
|---|---|---|---|---|---|---|---|---|
| Chinchilla (Hoffmann et al., 2022) | ● | | ● | ● | ● | | | |
| GLaM (Du et al., 2022) | | ◐ | | ● | ● | | | ● |
| GPT (Radford et al., 2018) | | | | | ● | | | |
| GPT-2 (Radford et al., 2019) | | ● | | | | | | |
| GPT-3 (Brown et al., 2020) | ● | ● | | ● | ● | | | ● |
| LaMDA (Thoppilan et al., 2022) | ● | | | ● | ● | | ◐ | |
| Llama 1 (Touvron et al., 2023a) | ● | | ● | ● | ● | ● | ● | |
| PaLM (Chowdhery et al., 2023) | | ◐ | | ● | ● | | | ● |
| Phi 2 (Abdin et al., 2023) | ● | | | ● | ● | ● | ● | |

## B    COUNTERARGUMENTS

We argue that AI-generated content from a variety of sources will be increasingly prevalent online, resulting in future genAI models being regularly trained on each other's outputs. We believe this vision of future data-mediated interactions between models is reasonable, based on evidence from academic literature and corporate reports. However, others may disagree with our argument. Here, we present possible counterarguments to our view to catalyze future work and discussion.

**Can't we use watermarks to filter AI-generated content from future internet-scraped datasets?** Several companies have publicly stated that they watermark AI-generated content (Dathathri et al., 2024; AI, 2024b; Clegg, 2024), making this argument plausible. Furthermore, (Drayson et al., 2025) show that using watermark detection techniques can help avoid model collapse under certain circumstances. However, reliance on watermarking has two major issues. First, watermarks are difficult to reliably detect and/or easily removed from generated content (Zhang et al., 2023; Longpre

Table 7: **Change in loss behavior for** $K = 2$ **interacting LLama models at various** $\alpha, \beta$. *We show results as* `initial` $\rightarrow$ `final` *prediction loss values for models on their own and the other models' tasks, at* `initial` $= T = 0$ *and* `final` $= T = 15$ *generations. For clarity, we colorize loss* increase *,* decrease *, and* constancy *($\Delta \leq 0.1$).*

| | $\beta = 0$ | | | $\beta = 0.5$ | | | $\beta = 1.0$ | | |
|---|---|---|---|---|---|---|---|---|---|
| | $\alpha = 0$ | $\alpha = 0.5$ | $\alpha = 1.0$ | $\alpha = 0$ | $\alpha = 0.5$ | $\alpha = 1.0$ | $\alpha = 0$ | $\alpha = 0.5$ | $\alpha = 1.0$ |
| Model 1 on Task 1 | $2.8 \rightarrow 4.5$ | $2.8 \rightarrow 3.2$ | $2.8 \rightarrow 3.1$ | $2.8 \rightarrow 2.9$ | $2.8 \rightarrow 2.9$ | $2.8 \rightarrow 3.0$ | $2.8 \rightarrow 2.8$ | $2.8 \rightarrow 2.8$ | $2.8 \rightarrow 3.2$ |
| Model 2 on Task 2 | $1.2 \rightarrow 2.0$ | $1.2 \rightarrow 1.9$ | $1.2 \rightarrow 1.8$ | $1.2 \rightarrow 1.2$ | $1.2 \rightarrow 1.3$ | $1.2 \rightarrow 1.9$ | $1.2 \rightarrow 1.2$ | $1.2 \rightarrow 1.2$ | $1.2 \rightarrow 1.8$ |
| Model 1 on Task 2 | $2.0 \rightarrow 2.9$ | $2.0 \rightarrow 1.9$ | $2.0 \rightarrow 1.7$ | $2.0 \rightarrow 2.5$ | $2.0 \rightarrow 1.4$ | $2.0 \rightarrow 1.8$ | $2.0 \rightarrow 2.6$ | $2.0 \rightarrow 1.3$ | $2.0 \rightarrow 1.7$ |
| Model 2 on Task 1 | $3.7 \rightarrow 5.0$ | $3.7 \rightarrow 3.1$ | $3.7 \rightarrow 3.1$ | $3.7 \rightarrow 4.2$ | $3.7 \rightarrow 3.0$ | $3.7 \rightarrow 3.0$ | $3.7 \rightarrow 4.4$ | $3.7 \rightarrow 2.9$ | $3.7 \rightarrow 3.1$ |

et al., 2024; Sadasivan et al., 2023). Second, detection via watermark requires sharing of watermark detection information, which is essentially a game-theoretic problem that relies on other companies' willingness to cooperate. Both issues make watermarks an unreliable mechanism to rid datasets of AI-generated content.

**What if there's less AI-generated online content than we think?** Some work suggests there may be less AI-generated content online than previously postulated (Matatov et al., 2024). However, other works consistently point to an uptick in AI-generated content online (Sun et al., 2025). We believe the widespread adoption and use of generative AI models across industries (AI, 2024a; Handa et al., 2025; Reuters, 2024), particularly for use in content creation (gen, 2022), provides strong evidence that AI-generated content will become a regular part of online life. Further empirical work is needed to vet both claims.

**What if one model provider dominates online content?** Although numerous generative AI models are available online, some evidence suggests that one or two companies may dominate the AI landscape. A market research firm estimated that in January 2025, Open AI's ChatGPT had 340 million monthly active users, Microsoft Copilot had 11 million, Google Gemini had 80 million, and Anthropic's Claude had 2 million (Zitron, 2025). If only one model/company dominates, our paradigm of training on other models' data will no longer be relevant and collapses back to the single model setting of prior work, e.g. (Shumailov et al., 2024; Kazdan et al., 2025; Dey & Donoho, 2024). Currently, though, this market research suggests that there are several models used by millions of users, making our assumptions somewhat reasonable. Future research could analyze market trends in model use and adoption to determine realistic assumptions.

## C  ADDITIONAL RESULTS

Tables 7 and 8 show full results for all tested $\beta, \alpha$ settings for the $K = 2$ setting across both the Llama and OPT model architectures. As stated in the main paper body, the $\beta = 0.5$ and $\alpha = 0.5$ settings perform best. Figures 4, 5, and 6 show loss values at each generation for (1) predicted theoretical results, (2) OPT $K = 2$ models, and (3) Llama $K = 2$ models. Finally, Figure 7 shows results for a $K = 3$ system of interacting OPT models.

## D  CODE FOR EXPERIMENTS

Code to generate the theory and experimental figures shown in the main paper body can be found at: https://github.com/arguslab-duke/multi_models_interaction.

## E  PROOFS FOR RESULTS IN SECTION 5

Before diving into the proofs, we recall the workflow setup and provide some preliminary results. The initial data are represented by matrix-vector pairs for the private data $(\tilde{X}_k, \tilde{y}_k), \ldots, (\tilde{X}_K, \tilde{y}_K)$

Table 8: **Change in loss behavior for $K = 2$ interacting OPT models at various $\alpha, \beta$** *We show results as* `initial` $\rightarrow$ `final` *prediction loss values for models on their own and the other models' tasks, at* `initial` $= T = 0$ *and* `final` $= T = 15$ *generations. For clarity, we colorize loss* `increase` , `decrease` , *and* `constancy` *($\Delta \leq 0.1$).*

|  | $\beta = 0$ | | | $\beta = 0.5$ | | | $\beta = 1.0$ | | |
|---|---|---|---|---|---|---|---|---|---|
|  | $\alpha = 0$ | $\alpha = 0.5$ | $\alpha = 1.0$ | $\alpha = 0$ | $\alpha = 0.5$ | $\alpha = 1.0$ | $\alpha = 0$ | $\alpha = 0.5$ | $\alpha = 1.0$ |
| Model 1 on Task 1 | $3.3 \rightarrow 4.8$ | $3.3 \rightarrow 3.6$ | $3.3 \rightarrow 3.5$ | $3.3 \rightarrow 3.3$ | $3.3 \rightarrow 3.3$ | $3.3 \rightarrow 3.5$ | $3.3 \rightarrow 3.1$ | $3.3 \rightarrow 3.2$ | $3.3 \rightarrow 3.5$ |
| Model 2 on Task 2 | $1.8 \rightarrow 2.6$ | $1.8 \rightarrow 2.2$ | $1.8 \rightarrow 2.1$ | $1.8 \rightarrow 1.7$ | $1.8 \rightarrow 1.7$ | $1.8 \rightarrow 2.2$ | $1.8 \rightarrow 1.5$ | $1.8 \rightarrow 1.8$ | $1.8 \rightarrow 2.2$ |
| Model 1 on Task 2 | $3.1 \rightarrow 3.7$ | $3.1 \rightarrow 2.2$ | $3.1 \rightarrow 2.2$ | $3.1 \rightarrow 3.5$ | $3.1 \rightarrow 1.8$ | $3.1 \rightarrow 2.2$ | $3.1 \rightarrow 3.4$ | $3.1 \rightarrow 1.8$ | $3.1 \rightarrow 2.2$ |
| Model 2 on Task 1 | $5.1 \rightarrow 5.6$ | $5.1 \rightarrow 3.6$ | $5.1 \rightarrow 3.5$ | $5.1 \rightarrow 5.1$ | $5.1 \rightarrow 3.5$ | $5.1 \rightarrow 3.5$ | $5.1 \rightarrow 5.4$ | $5.1 \rightarrow 3.5$ | $5.1 \rightarrow 3.4$ |

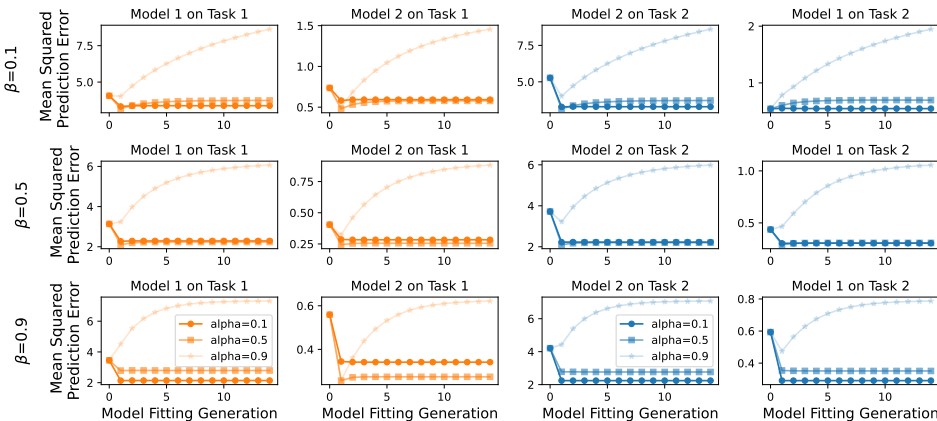

Figure 4: **Predicted behavior over time for a $K = 2$ model system with varying $\alpha, \beta$.** *We use equations for MSE from theorem 2 and run simulations with $K = 2$, dimension 50, rank 15.*

and for the public data $(X_*, y_*)$. At each generation $t$, the data $(X_{tk}, y_{tk})$ produced by entity $k$ is given by

$$y_{tk} = X_{tk}\hat{\theta}_{t-1,k} + w_{tk}$$

where $\hat{\theta}_{t-1,k} \in \mathbb{R}^d$ is the most recent parameter estimate and $w_{tk} \sim \mathsf{N}(0, \sigma^2\mathrm{I})$ is Gaussian noise that is independent across entities and generations. To represent the notation compactly, we use the embedding

$$\tilde{\boldsymbol{X}} = \begin{bmatrix} \tilde{X}_1 & & \\ & \ddots & \\ & & \tilde{X}_K \end{bmatrix}, \quad \boldsymbol{y}_0 = \begin{bmatrix} \tilde{y}_1 \\ \vdots \\ \tilde{y}_K \end{bmatrix} \quad \boldsymbol{X}_t = \begin{bmatrix} X_{t1} & & \\ & \ddots & \\ & & X_{tK} \end{bmatrix}, \quad \boldsymbol{y}_t = \begin{bmatrix} y_{t1} \\ \vdots \\ y_{tK} \end{bmatrix}, \quad \boldsymbol{w}_t = \begin{bmatrix} w_{t1} \\ \vdots \\ w_{tK} \end{bmatrix},$$

**Linear dynamical system** According to the workflow, each parameter estimate is obtained as the minimizer in $\theta \in \mathbb{R}^d$ of the empirical loss:

$$\bar{\alpha}_t\beta_t\|\tilde{y}_k - \tilde{X}_k\theta\|^2 + \bar{\alpha}_t\bar{\beta}_t\|y_* - X_*\theta\|^2 + \frac{\alpha_t}{K}\sum_{j=1}^{K}\|y_{tj} - X_{tj}\theta\|^2,$$

where $\alpha_0 \equiv 0$ and so only the first two terms are present at initialization. The minimum norm solution is given in closed form by

$$\hat{\theta}_{tk} = \left(\bar{\alpha}_t\beta_t\tilde{S}_k + \bar{\alpha}_t\bar{\beta}_tS_*^\top + \frac{\alpha_t}{K}\sum_{j=1}^{k}S_{tk}\right)^+ \left(\bar{\alpha}_t\beta_t\tilde{X}_k^\top\tilde{y}_t + \bar{\alpha}_t\bar{\beta}_tX_*^\top y_* + \frac{\alpha_t}{K}\sum_{j=1}^{k}X_{tj}^\top y_{tj}\right)$$

Evaluation loss across generations for different $\alpha$ and $\beta$ values (K=2) OPT-350m

Figure 5: **Actual behavior over time for interactions between OPT models** ($K = 2$) **with varying** $\alpha, \beta$.

Evaluation loss across generations for different $\alpha$ and $\beta$ values (K=2) Llama-3.2-1B

Figure 6: **Actual behavior over time for interactions between Llama models** ($K = 2$) **with varying** $\alpha, \beta$.

where $\tilde{S}_k = \tilde{X}_t^\top \tilde{X}_k$, $S_* = X_t^\top X_*$, $S_{tk} = X_{tk}^\top X_{tk}$, and $(\cdot)^+$ denotes the Moore-Penrose pseudoinverse. Stacking the estimates into a vector $\hat{\boldsymbol{\theta}} = \text{vec}(\hat{\theta}_1, \dots, \hat{\theta}_K)$ and using the identity $A^\top = A^\top A A^+$, we can express all estimate updates simultaneously as

$$\hat{\boldsymbol{\theta}}_t = \boldsymbol{G}_t^+ \left( \bar{\alpha}_t \beta_t \tilde{\boldsymbol{S}} \tilde{\boldsymbol{X}}^+ \tilde{\boldsymbol{y}} + \bar{\alpha}_t \bar{\beta}_t (\mathbf{1}_K \otimes S_*) X_*^+ y_* + \frac{\alpha_t}{K} (\mathbf{1}_K \otimes \sum_{j=1}^k S_{tj} X_{tj}^+ y_{tj}) \right), \qquad (3)$$

where $\mathbf{1}_K$ denotes the $K \times 1$ vector of ones and $\boldsymbol{G}_t$ is the block diagonal matrix given by

$$\boldsymbol{G}_t := \bar{\alpha}_t \beta_t \tilde{\boldsymbol{S}} + \bar{\alpha}_t \bar{\beta}_t (\mathrm{I}_K \otimes S_*) + \alpha_t (\mathrm{I}_K \otimes \underline{\boldsymbol{S}}_t), \qquad \underline{\boldsymbol{S}}_t = \frac{1}{K} \sum_{k=1}^K S_{kt}.$$

Defining the orthogonal projection matrix $\Pi = \frac{1}{K} \mathbf{1}_K \mathbf{1}_K^\top \otimes \mathrm{I}_d$ we can write

$$\frac{1}{K} (\mathbf{1}_K \otimes \sum_{j=1}^k S_{tj} X_{tj}^+ y_{tj}) = \Pi \boldsymbol{S}_t \boldsymbol{X}_t^+ \boldsymbol{y}_t.$$

Introducing the matrices $\boldsymbol{P}_t := \bar{\alpha}_t \boldsymbol{G}_t^+ \begin{bmatrix} \beta_t \tilde{\boldsymbol{S}} & \bar{\beta}_t (\mathbf{1}_K \otimes S_*) \end{bmatrix}$ and $\boldsymbol{Q}_t := \alpha_t \boldsymbol{G}_t^+ \Pi \boldsymbol{S}_t$, can express (3) as

$$\hat{\boldsymbol{\theta}}_t = \boldsymbol{P}_t \begin{bmatrix} \tilde{\boldsymbol{X}}^+ \tilde{\boldsymbol{y}} \\ X_*^+ y_* \end{bmatrix} + \boldsymbol{Q}_t \boldsymbol{X}_t^+ \boldsymbol{y}_t.$$

Using $\boldsymbol{y}_t = \boldsymbol{X}_t \boldsymbol{\theta}_{t-1} + \boldsymbol{w}_t$ and $\boldsymbol{Q}_t \boldsymbol{X}_t^+ \boldsymbol{X}_t = \boldsymbol{Q}_t$, we obtain

$$\hat{\boldsymbol{\theta}}_t = \boldsymbol{P}_t \begin{bmatrix} \tilde{\boldsymbol{X}}^+ \tilde{\boldsymbol{y}} \\ X_*^+ y_* \end{bmatrix} + \boldsymbol{Q}_t \hat{\boldsymbol{\theta}}_{t-1} + \boldsymbol{Q}_t \boldsymbol{X}_t^+ \boldsymbol{w}_t. \qquad (4)$$

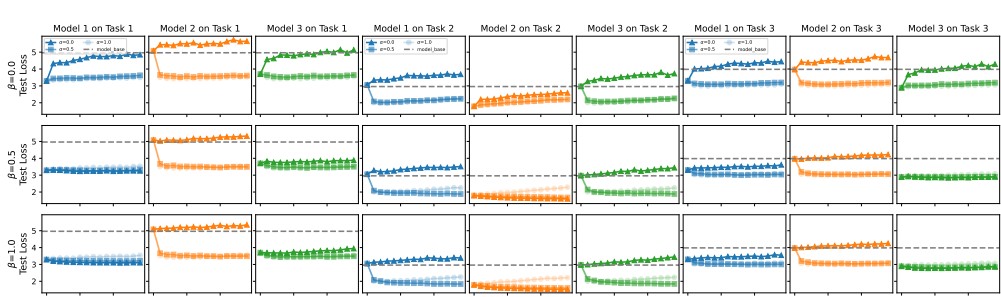

Figure 7: **Actual behavior over time for interactions between OPT models ($K = 3$) with varying $\alpha, \beta$.**

| Parameter | Value |
|---|---|
| bf16 | True |
| data_seed | generation |
| dataloader_num_workers | 4 |
| dataset_text_field | input_ids |
| eval_on_start | True |
| eval_strategy | steps |
| eval_steps | 100 |
| gradient_accumulation_steps | 2 |
| gradient_checkpointing | False |
| include_num_input_tokens_seen | True |
| learning_rate | $8 \times 10^{-6}$ |
| logging_steps | 5 |
| lr_scheduler_type | constant_with_warmup |
| max_seq_length | 512 |
| max_steps | 100 |
| num_train_epochs | 1 |
| optim | adamw_torch |
| per_device_eval_batch_size | 16 |
| per_device_train_batch_size | 2 (8 for T=8 experiments) |
| remove_unused_columns | False |
| seed | generation |
| warmup_ratio | 0.025 |
| save_strategy | no |

Table 9: Fine-tuning hyperparameters used.

This expression shows that the estimates evolve according to a discrete-time linear dynamical system (also known as a Kalman filter model) with state variable $\hat{\boldsymbol{\theta}}_t$.

### E.1 PROOF OF THEOREM 1

We now derive the distribution of $\hat{\boldsymbol{\theta}}_t$ conditional on the initial data $D_0$ given by $(\tilde{\boldsymbol{X}}, \tilde{\boldsymbol{y}})$ and $(X_*, y_*)$. Specifically, we show that the estimates are Gaussian with mean and variance

$$\mathbb{E}[\hat{\boldsymbol{\theta}}_t \mid D_0] = \boldsymbol{M}_t \begin{bmatrix} \tilde{\boldsymbol{X}}^+ \tilde{\boldsymbol{y}} \\ X_*^+ y_* \end{bmatrix}, \qquad \mathsf{Cov}(\hat{\boldsymbol{\theta}}_t \mid D_0) = \boldsymbol{C}_t$$

where the matrices $\boldsymbol{M}_t$ and $\boldsymbol{C}_t$ are defined recursively with $\boldsymbol{M}_0 = \boldsymbol{P}_0$ and $\boldsymbol{C}_0 = \boldsymbol{0}_{Kd \times Kd}$ and

$$\boldsymbol{M}_t = \boldsymbol{P}_t + \boldsymbol{Q}_t \boldsymbol{M}_{t-1}, \qquad \boldsymbol{C}_t = \boldsymbol{Q}_t(\sigma^2 \boldsymbol{S}_t^+ + \boldsymbol{C}_{t-1})\boldsymbol{Q}_t, \qquad t \geq 1.$$

The proof is by mathematical induction. For the base case $t = 0$ we invoke (4) along with $\boldsymbol{Q}_0 = 0$ to see that $\hat{\boldsymbol{\theta}}_0$ is a deterministic function of the initial data with $\boldsymbol{M}_0 = \boldsymbol{P}_0$ and $\boldsymbol{C}_0 = \boldsymbol{0}_{Kd \times Kd}$.

For the inductive case, assume that the stated distribution holds up to generation $t - 1$. From the definition of the workflow, (4) holds with $\boldsymbol{w}_t \sim \mathsf{N}(0, \sigma^2 \mathrm{I})$ independent of everything else. Thus $\hat{\boldsymbol{\theta}}_t$ is Gaussian with mean

$$\mathbb{E}[\hat{\boldsymbol{\theta}}_t \mid D_0] = \boldsymbol{P}_t \begin{bmatrix} \tilde{\boldsymbol{X}}^+ \tilde{\boldsymbol{y}} \\ X_*^+ y_* \end{bmatrix} + \boldsymbol{Q}_t \mathbb{E}[\hat{\boldsymbol{\theta}}_{t-1} \mid D_0] = \underbrace{(\boldsymbol{P}_t + \boldsymbol{Q}_t \boldsymbol{M}_{t-1})}_{\boldsymbol{M}_t} \begin{bmatrix} \tilde{\boldsymbol{X}}^+ \tilde{\boldsymbol{y}} \\ X_*^+ y_* \end{bmatrix}$$

and covariance

$$\mathsf{Cov}(\hat{\boldsymbol{\theta}}_t \mid D_0) = \mathsf{Cov}(\boldsymbol{Q}_t \hat{\boldsymbol{\theta}}_{t-1} \mid D_0) + \mathsf{Cov}(\boldsymbol{Q}_t X_t^+ w) = \underbrace{\boldsymbol{Q}_t C_{t-1} \boldsymbol{Q}_t + \sigma^2 \boldsymbol{Q}_t \boldsymbol{S}_t^+ \boldsymbol{Q}_t}_{C_t}.$$

This concludes the proof of Theorem 1. □

### E.2 PROOF OF THEOREM 2

Under the assumptions of the theorem, we have that

$$\begin{bmatrix} \tilde{\boldsymbol{X}}^+ \tilde{\boldsymbol{y}} \\ X_*^+ y_* \end{bmatrix} \sim \mathsf{N}\left( \begin{bmatrix} \tilde{\boldsymbol{S}} \tilde{\boldsymbol{S}}^+ & 0 \\ 0 & S_* S_*^+ \end{bmatrix} (\boldsymbol{1}_{K+1} \otimes \theta), \sigma^2 \begin{bmatrix} \tilde{\boldsymbol{S}}^+ & 0 \\ 0 & S_*^+ \end{bmatrix} \right). \tag{5}$$

The goal for this proof is to verify that if $\boldsymbol{G}_1, \ldots, \boldsymbol{G}_t$ are full rank then

$$\mathbb{E}[\hat{\boldsymbol{\theta}}_t] = \big(\mathrm{I} - \boldsymbol{Q}_t \cdots \boldsymbol{Q}_1(\mathrm{I} - \boldsymbol{G}_0 \boldsymbol{G}_0^+)\big)(\boldsymbol{1}_K \otimes \theta), \quad \mathsf{Cov}(\hat{\boldsymbol{\theta}}_t) = \boldsymbol{M}_t \begin{bmatrix} \tilde{\boldsymbol{S}}^+ & 0 \\ 0 & S_*^+ \end{bmatrix} \boldsymbol{M}_t^\top + C_t.$$

We proceed by mathematical induction. Consider the case $t = 0$. By (4) along with $\boldsymbol{Q}_0 = 0$, the mean is

$$\mathbb{E}[\boldsymbol{\theta}_0] = \boldsymbol{P}_t \begin{bmatrix} \tilde{\boldsymbol{S}} \tilde{\boldsymbol{S}}^+ & 0 \\ 0 & S_* S_*^+ \end{bmatrix} (\boldsymbol{1}_{K+1} \otimes \theta) = \boldsymbol{G}_0 \boldsymbol{G}_0^+ (\boldsymbol{1}_K \otimes \theta).$$

Likewise, recalling that $\boldsymbol{M}_0 = \boldsymbol{P}_0$, the variance is

$$\mathsf{Cov}(\boldsymbol{\theta}_0) = \boldsymbol{M}_0 \, \mathsf{Cov}\left( \begin{bmatrix} \tilde{\boldsymbol{X}}^+ \tilde{\boldsymbol{y}} \\ X_*^+ y_* \end{bmatrix} \right) \boldsymbol{M}_0^\top = \sigma^2 \boldsymbol{M}_0 \begin{bmatrix} \tilde{\boldsymbol{S}}^+ & 0 \\ 0 & S_*^+ \end{bmatrix} \boldsymbol{M}_0^\top.$$

Next, suppose that $\boldsymbol{G}_1, \ldots, \boldsymbol{G}_t$ are full rank and the stated distribution holds up to time $t - 1$. By Theorem 1 we know that $\hat{\boldsymbol{\theta}}_t$ is Gaussian and so all that remains is to verify the given expressions for the mean and covariance. By the linearity of expectation and (4), the mean satisfies

$$\begin{aligned} \mathbb{E}[\hat{\boldsymbol{\theta}}_t] &= \boldsymbol{P}_t \begin{bmatrix} \mathbb{E}[\tilde{\boldsymbol{X}}^+ \tilde{\boldsymbol{y}}] \\ \mathbb{E}[X_*^+ y_*] \end{bmatrix} + \boldsymbol{Q}_t \mathbb{E}[\hat{\boldsymbol{\theta}}_{t-1}] \\ &= \boldsymbol{P}_t(\mathrm{I}_{K+1} \otimes \theta) + \boldsymbol{Q}_t(\mathrm{I}_K \otimes \theta) - \boldsymbol{Q}_t \boldsymbol{Q}_{t-1} \cdots \boldsymbol{Q}_1(\mathrm{I} - \boldsymbol{G}_0 \boldsymbol{G}_0^+)(\boldsymbol{1}_K \otimes \theta), \end{aligned}$$

where the last follows from the inductive assumption applied to $\mathbb{E}[\hat{\boldsymbol{\theta}}_{t-1}]$. Moreover, from the definitions of $\boldsymbol{P}_t$ and $\boldsymbol{Q}_t$, we have

$$\begin{aligned} \boldsymbol{P}_t(\mathrm{I}_{K+1} \otimes \theta) + \boldsymbol{Q}_t(\mathrm{I}_K \otimes \theta) &= \boldsymbol{G}_t^+\big(\bar{\alpha}_t \beta_t \tilde{\boldsymbol{S}} + \bar{\alpha}_t \bar{\beta}_t(\mathrm{I}_K \otimes S_*)\big) + \alpha_t \Pi \boldsymbol{S}_t\big)(\mathrm{I}_K \otimes \theta) \\ &= \boldsymbol{G}_t^+ \boldsymbol{G}_t(\mathrm{I}_K \otimes \theta) \\ &= \mathrm{I}_K \otimes \theta, \end{aligned}$$

where the second line follows from the identity $\Pi \boldsymbol{S}_t(\boldsymbol{1}_K \otimes \mathrm{I}_d) = (\mathrm{I}_K \otimes \underline{\boldsymbol{S}}_t)(\boldsymbol{1}_K \otimes \mathrm{I}_d)$ and the last line holds because $\boldsymbol{G}_t$ is full rank. Combining the above displays gives the desired expression for the mean. The expression for the variance follows directly from (5) and Theorem 1. □

### E.3 PROOF OF THEOREM 3

If the spectral radius of $\boldsymbol{Q}$ is strictly less than one, then $\boldsymbol{Q}^t \to \boldsymbol{0}$ as $t \to \infty$, and the Neumann series in (1) converge to the well-defined limits in (2). These limits can also be seen as the (necessarily unique) solutions to the fixed point equations

$$\boldsymbol{M} = \boldsymbol{P} + \boldsymbol{M}\boldsymbol{Q}, \qquad \boldsymbol{C} = \boldsymbol{Q}(\boldsymbol{C} + \sigma^2 \boldsymbol{S}^+)\boldsymbol{Q}^\top,$$

where the expression for the covariance is known as the discrete time Lyapunov equation. Combining these convergence results with Theorem 2 completes the proof. □.

### E.4 PROOF OF LEMMA 1

If $\alpha = 0$ or if $\boldsymbol{S} = \boldsymbol{0}$ then $\boldsymbol{Q} = \boldsymbol{0}$ and so the stated result holds. Henceforth, we assume $0 < \alpha < 1$ and $\boldsymbol{S}$ is nonzero. Suppose that $\gamma \boldsymbol{S} = \lambda \tilde{\boldsymbol{S}} + (1 - \lambda)(\mathrm{I}_K \otimes S_*)$ for some $0 < \beta \le \lambda \le 1$ and $\gamma > 0$. Then,

$$\begin{aligned}
\boldsymbol{G} &= \bar{\alpha}\beta\tilde{\boldsymbol{S}} + \bar{\alpha}\bar{\beta}(\mathrm{I}_K \otimes S_*) + \alpha(\mathrm{I}_K \otimes \underline{\boldsymbol{S}}) \\
&= \frac{\bar{\alpha}\beta}{\lambda}(\gamma \boldsymbol{S} - (1-\lambda)(\mathrm{I}_K \otimes S_*)) + \bar{\alpha}\bar{\beta}(\mathrm{I}_K \otimes S_*) + \alpha(\mathrm{I}_K \otimes \underline{\boldsymbol{S}}) \\
&= \frac{\bar{\alpha}\beta\gamma}{\lambda}\boldsymbol{S} + \bar{\alpha}\Big(\frac{\lambda - \beta}{\lambda}\Big)(\mathrm{I}_K \otimes S_*) + \alpha(\mathrm{I}_K \otimes \underline{\boldsymbol{S}}).
\end{aligned}$$

Hence,

$$\boldsymbol{Q} = \big(\delta\boldsymbol{S} + \mathrm{I}_K \otimes \Delta\big)^+ \Pi\boldsymbol{S}, \qquad \delta = \frac{\bar{\alpha}\beta\gamma}{\alpha\lambda}, \qquad \Delta = \frac{\lambda - \beta}{\alpha\lambda}S_* + \underline{\boldsymbol{S}}.$$

To proceed, observe that each diagonal block of $\boldsymbol{S} = \mathrm{diag}(S_1, \ldots, S_K)$ lies in the span of $\underline{\boldsymbol{S}} = \frac{1}{K}\sum_{k=1}^{K} S_k$, and thus $\boldsymbol{S}$ lies in the span of $\mathrm{I}_K \otimes \Delta$. Accordingly, we can write

$$\boldsymbol{S}\big(\delta\boldsymbol{S} + \mathrm{I}_K \otimes \Delta\big)^+ = (\mathrm{I}_K \otimes \Delta^{1/2})\boldsymbol{R}\big(\delta\boldsymbol{R} + \mathrm{I}\big)^{-1}(\mathrm{I}_K \otimes \Delta^{+/2}),$$

where $(\cdot)^{1/2}$ denote the symmetric positive semidefinite square root of a positive semidefinite and $\boldsymbol{R} \coloneqq (\mathrm{I}_K \otimes \Delta^{+/2})\boldsymbol{S}(\mathrm{I}_K \otimes \Delta^{+/2})$. To bound the spectral radius, denoted by $\rho(\cdot)$, we use that fact that the eigenvalues of $AB$ and $BA$ are the same for any square matrices $A$ and $B$ along with that fact that $\mathrm{I}_K \otimes \Delta$ commutes with $\Pi$ to write

$$\begin{aligned}
\rho(\boldsymbol{Q}) &= \rho\Big(\big(\delta\boldsymbol{S} + \mathrm{I}_K \otimes \Delta\big)^+ \Pi\boldsymbol{S}\Big) \\
&= \rho\Big(\boldsymbol{S}\big(\delta\boldsymbol{S} + \mathrm{I}_K \otimes \Delta\big)^+ \Pi\Big) \\
&= \rho\Big((\mathrm{I}_K \otimes \Delta^{1/2})\boldsymbol{R}\big(\delta\boldsymbol{R} + \mathrm{I}\big)^{-1}(\mathrm{I}_K \otimes \Delta^{+/2})\Pi\Big) \\
&= \rho\Big(\Pi\boldsymbol{R}\big(\delta\boldsymbol{R} + \mathrm{I}\big)^{-1}\Pi\Big) \\
&= \|\Pi\boldsymbol{R}\big(\delta\boldsymbol{R} + \mathrm{I}\big)^{-1}\Pi\|,
\end{aligned}$$

where $\|\cdot\|$ denotes the operator norm and the last equality holds because $\Pi\boldsymbol{R}\big(\delta\boldsymbol{R} + \mathrm{I}\big)^{-1}\Pi$ is symmetric positive semidefinite. Letting $\epsilon > 0$ denote the smallest nonzero singular value of $\boldsymbol{R}$, we have

$$\begin{aligned}
(1 + \epsilon\delta)\rho(\boldsymbol{Q}) &\le \|\Pi\boldsymbol{R}\Pi\| \\
&\overset{(a)}{=} \|\Pi(\mathrm{I}_K + \Delta^{+/2})\boldsymbol{S}(\mathrm{I}_K \otimes \Delta^{+/2})\Pi\| \\
&\overset{(b)}{=} \|(\mathrm{I}_K \otimes \Delta^{+/2})\Pi\boldsymbol{S}\Pi(\mathrm{I}_K \otimes \Delta^{+/2})\| \\
&\overset{(c)}{=} \|(\mathrm{I}_K \otimes \Delta^{+/2})(\tfrac{1}{K}\mathbf{1}_{K\times K} \otimes \underline{\boldsymbol{S}})(\mathrm{I}_K \otimes \Delta^{+/2})\| \\
&\overset{(d)}{=} \|\tfrac{1}{K}\mathbf{1}_{K\times K} \otimes \Delta^{+/2}\underline{\boldsymbol{S}}\Delta^{+/2}\| \\
&\overset{(e)}{=} \|\tfrac{1}{K}\mathbf{1}_{K\times K}\|\|\Delta^{+/2}\underline{\boldsymbol{S}}\Delta^{+/2}\| \\
&\overset{(f)}{=} \|\Delta^{+/2}\underline{\boldsymbol{S}}\Delta^{+/2}\|
\end{aligned}$$

where (a) is the definition of $\boldsymbol{R}$; (b) follows from the commutativity of $\Pi$ and $\mathrm{I}_K \otimes \Delta^{+/2}$; (c) follows from $\Pi\boldsymbol{S}\Pi = \frac{1}{K}\mathbf{1}_{K\times K} \otimes \underline{\boldsymbol{S}}$; (d) is the mixed-product property of the Kronneker product; (e) is the basic identity $\|A \otimes B\| = \|A\|\|B\|$ for any matrices $A$ and $B$; and (f) holds because $\|\frac{1}{K}\mathbf{1}_{K\times K}\| = 1$. Finally, using that $0 \preceq \underline{\boldsymbol{S}} \preceq \Delta$, we see that $\|\Delta^{+/2}\underline{\boldsymbol{S}}\Delta^{+/2}\| \le \|\Delta^{+/2}\Delta\Delta^{+/2}\| \le 1$. This verifies that $\rho(\boldsymbol{Q})$ is strictly less than one. $\qquad\square$

