# OpenReview forum: "What happens when generative AI models train recursively on each others' outputs?"
_ICLR.cc/2026/Conference — ICLR 2026 Poster_

### Official Review · Reviewer_xD3M · 2025-10-19

**Soundness:** 3
**Presentation:** 2
**Contribution:** 3
**Rating:** 6
**Confidence:** 3

**Summary:**

The paper describes work on investigating effects of recursive training of model generated data (+ combinations of human authored data) with LLMs. The authors draw motivation for the need for research in this direction by highlighting several realities and overlooked realities such as proprietary LLMs are trained mostly with internet-scraped data and how these datasets overlap substantially. The most important motivation that the authors emphasize is that future LLMs will evidently be trained will LLM-generated (their own) data. The authors back this with existing literature from learning theory on model collapse. In order to further shed light on the potential good or bad consequences of this model-data interaction across tasks, the authors formalize an interactive training pipeline controlled by alpha (fraction of new data per iteration) and beta (public data-private data partition). Results show from a small-scale experiment of two model providers (K=2) derived from Llama and OPT model architectures with t = 15 show that setting alpha and beta to 0.5 (equal partition) seemingly optimal results across tasks (science QA and math QA) compared to other values. Setting alpha to 0 denoting use of purely human-generated data results to degradation in task generalization while setting this to 1 denoting purely LLM-generated data results to equal degradation in the original task.

Overall, the paper does present a simple and understandable method for potentially simulating model collapse and interactions upon training from LLM-generated data but my main issue is centered on grounding experiments with more rigor such as exploring K=10/20/50 or t=50. See feedback below for other concerns with the paper.

**Strengths:**

The paper proposes a simple yet intuitive method for exploring training data dynamics of LLMs as shown in Figure 1. I found the paper to be fairly readable and the way the paper motivates the need for investigating recursive training from model-generated data to be useful in contextualizing the study and support for the experiments. I believe the model collapse research community may find this paper's results to be beneficial and interesting.

**Weaknesses:**

While I appreciate the simplicity and readability of the paper, there are some issues that I found that the authors can use to improve the quality/rigor of the study:

First, the current experiment setup seems shortsighted to me with using only two model providers of K=2. Likewise, in terms of iteration, a realistic scenario would be repeatedly training on model-generated data by a longer margin, say t = 30 or 50 or even 100. The goal is to rigorously investigate how far can the performance converge or if there are possibilities of similar phenomenon like grokking. Model providers these days are extremely fast in releases (almost a new one every 3-4 months), hence I believe a more realistic setup is needed for the study. Likewise, a larger K such as 10 could also be explored to further investigate effects of training from a diverse collection of models.

I believe the paper lacks equal discussion on the implications of training data dynamics that are grounded from the results. For example, the study shows that using a perfect split of 0.5 for alpha and beta seems to produce optimal results but this seems very idealistic and tied to the current experiment setup. More realistic setups might be more nuanced and highly dependent on factors such as training data quality, task diversity, etc. How can the study account for this? This part is underdeveloped in the paper.

I suggest the authors to balance the structure of the paper by prioritizing experiment results. The current paper’s experiments and discussion are both pushed back to the last 2 pages of the paper and feels quite rushed/limited when you read it. The first four pages motivating the challenge could be condensed further to prioritize the results. I would also appreciate more expanded and clear discussion on task generalization as well as this seems underdeveloped as well.

Please improve your references and cite published articles. In the introduction alone, the main citations to support interdisciplinary use case adoption of generative AI are mostly blogs and websites, ignoring more qualified literature or previous works that have been peer-reviewed.

**Questions:**

“While some amount of mixing improves model performance on previously-unseen tasks, homogenization occurs for D∗ at all α and for  ̃Dk when α > 0—everywhere it can” - this sentence is confusing, can you please clarify and expand this?

---

> ### Author Response · Authors · 2025-11-22
>
> We thank the reviewer for their helpful feedback. We've extensively revised the paper to address the reviewer's concerns, and we point the reviewer to relevant revised sections on our response below.
>
> - **Paper structure / priority of experiments**.
>
>  We thank the reviewer for pointing this out! In response to your notes, we have condensed the practical setup explanations in Section 3 and have moved  tables from Section 3 to the Appendix. We've also added rationale for the linear simplification earlier in the introduction, as well as in the theory section. This allows us to dive into the ``meat'' of the paper earlier, around the bottom of page 3.
>
> - **Small K / short T in experiments**.
>
> We agree the limited $K$ and $T$ in our experimental evaluation limit understanding of how our result might scale. During the revision period, we've run experiments with $K=3$ models (for shorter $T=8$), and we plan to run experiments with larger $T$ soon. The updated results obtained so far are presented in column (c) in Table 2 and Table 4 of the revision. Unfortunately, due to the time needed to generate data for the next training generation, it takes a long time to train each generation of models (about 18 hours per generation on our compute cluster when $\alpha = 1.0$). We can likely complete a run with larger $T \approx 20$ before the camera-ready deadline, but any more than that just takes too long in the $\alpha=1.0$ scenario for large $K$. Unfortunately, $K=10$ is simply not feasible on our current compute setup for large state of the art LLMs. We have noted it in the paper as important future work.
>
> If the reviewer is interested, we have expanded our experimental evaluations by adding current results on K=3 for Llama 3.2 1B models and larger Llama 3.2 3B models for the $K=2$ settings. New experimental results can be found in Table 2 column (c) and Table 4. Table 2 (c) shows the results for interactions between k=2 larger llm models (Llama 3.2 3B) at $\beta=0.5$. Table 4. shows new results for extending the Llama 3.2 1B result from $K=2 \rightarrow K=3$ at $\beta=0.5$ and comparing it with our results for $K=3$ smaller OPT3 models. We also updated our experimental setup section to explain the task specific dataset for the third company being AI2 ARC dataset of science reasoning questions. Our additional results echo previous finding that cross-model training can both diversify and homogenize generative models as we scale up model size and including diversity by increasing $K$.
>
> - **Reference quality**.
>
> Again, thank you for pointing this out. In the revised paper, we have made a concerted effort to include relevant peer-reviewed sources and to reduce reliance on informal references, particularly in the introduction. If the reviewer has suggestions for additional relevant work that we may have overlooked, we would greatly appreciate them.
>
> - **Confusing sentence about homogenization**.
>
> We apologize for the unclear wording. What we were trying to say is that the mixing in our model setup leads models to behave more homogeneously (e.g. as show in the PCA plots, feature representations become more similar). For $D^{\*}$,  the common dataset, this homogenization occurs consistently. For $\\tilde{D}_k$, this occurs for $\alpha > 0$, which is the setting when models actually transfer information about their private datasets to one another via the mixing. That's why we added the phrase "everywhere it can," since under all these settings ($D^*$ for all $\alpha$ and $\tilde{D}_k$ with $\alpha >0$), models are sharing underlying dataset information. We've attempted to explain this concept better in the paper -- see Section 6.

---

> > ### Comment · Reviewer_xD3M · 2025-11-25
> > **Acknowledging authors' response**
> >
> > This is to confirm that I have read the authors' response to my review as well as my co-reviewers' feedback and corresponding responses by the authors.
> >
> > Thanks to the authors for clarifying my concerns and conducting additional experiments with a slightly larger K for Llama3.2. If it is computationally infeasible to simulate values for K that are >10 and T that are >50, please make it clear in the paper. Upon reading the paper, this is the question that immediately came to mind on why the authors did not explore increased values for deeper insights. Nonetheless, this is still an interesting study.

---

> > > ### Author Response · Authors · 2025-11-26
> > >
> > > We thank the reviewer for the positive comments and helpful feedback. As suggested, we will explicitly note the computational limits of scaling the experiments. Specifically, we will add the following sentence:
> > >
> > > “Although our framework can, in principle, accommodate an arbitrary number of interacting entities, exploring configurations with large K and T was computationally infeasible, as each setting requires repeated fine-tuning and evaluation of multiple large models. We therefore restricted our experiments to smaller scales that preserve interpretability while remaining practically achievable.”

---

### Official Review · Reviewer_1aha · 2025-10-20

**Soundness:** 4
**Presentation:** 3
**Contribution:** 2
**Rating:** 6
**Confidence:** 4

**Summary:**

The authors consider what happens when a collection of models is iteratively trained on a combination of public data, private data, and data generated by all the other models. They begin by arguing that this setup reflects reality by surveying training datasets for a variety of generative AI models. Then, they present a simplified linear regression model for this setting. They derive the bias and variance of the collection of models after $t$ training iterations. They find that the models all converge most efficiently when about half of their initial data is public and about half of their data in future iterations consists of prior generation outputs. This prediction is validated in experiments training OPT and Llama, where each one has a private dataset (SciQ or GSM8k). Models are able to do transfer learning from each others' outputs.

**Strengths:**

- The overall paper, presentation, and writing quality is high. This is a very well-executed research project.
- The research topic of recursive training dynamics with multiple models is important, timely, and interesting
- The model is well-designed (Figure 1 and Section 4 are great; I wish they had come two pages sooner)
- The experimental setup is clever

**Weaknesses:**

- What we learn in the multi-model setting is limited and follows expectations: just as we've seen in single-model collapse with accumulation, but different private data can lead to some transfer learning in the population
- The paper could do a better job of providing the intuitive takeaways from the theorems.
- The paper is fairly verbose and repetitive; the core of the paper doesn't begin until page 5


### Overall evaluation
This is a tricky paper to evaluate, as it's a very high-quality paper, but what we learn feels limited. Perhaps other reviewers will feel differently. This paper definitely deserves to be published and does contribute to the area of model collapse. The structure of the paper could be improved, getting to the contributions more quickly and providing clearer takeaways from the theory.

**Questions:**

1. What exactly are the takeaways from the theoretical analysis? We see that under certain conditions, each model's estimate converges to the true parameter. Is Figure 2 the real takeaway from this section?
2. In addition to loss, were there also the same patterns in model accuracy on SciQ and GSM8k?

### Comments
- It's a bit jarring for the paper to go from discussing generative AI for so long and then jump to a linear regression model. The rationale for this simplification makes sense, but should be mentioned/justified earlier in the abstract/intro (e.g., in the context of related work).
- Section 3 has a lot of text, references, and tables for some well-known facts about model training. It could be summarized in a paragraph
- the motivation in the intro for why it matters to study recursive training among multiple models vs just one single model is lacking. The intro just says it has "received little attention," but doesn't explain why it matters that there are multiple models rather than just one. Are there new and different dynamics that occur? Otherwise we could just assume it's basically the same and that studying a single model recursively training is sufficient to understand what happens with multiple.
- Some of the intro and related work was repetitive.
- some in-text citations missing an author (gen, 2022), (app, 2024)
- from the related work, it wasn't clear what it means for all models to have a "bound in error $\pi^2/6$"
- Line 100: "long term effects" is unclear; what is meant is iterative training dynamics, right?
- Line 157: ... this doesn't seem overlooked. It seems like a widely understood fact that many models use CommonCrawl, arXiv, GitHub, Wikipedia, etc
- Figure 1 is very nice.

---

> ### Author Response · Authors · 2025-11-22
>
> We thank the reviewer for their helpful feedback and respond to their points below. We have revised our paper to address the reviewer's concerns and will point them to appropriate revisions throughout our response. Revised paper text is teal.
>
> - **Presentation and structure suggestions**.
>
> We thank the reviewer for pointing this out! In response to your notes, we have condensed the practical setup explanations in Section 3 and have moved  tables from Section 3 to the Appendix. We've also added rationale for the linear simplification earlier in the introduction, as well as in the theory section. This allows us to dive into the ``meat'' of the paper earlier, around the bottom of page 3.
>
> - **Limited new dynamics beyond single-model collapse**.
>
> We appreciate this perspective but respectfully disagree. While we do build on work from the single-model collapse literature, our multi-model setting is distinct. In this setup, models can transfer beneficial private information to others (positive externalities) while also homogenizing behaviors across providers (negative externalities). We've added a "key insights" paragraph in the introduction and discussion to better emphasize this contribution, and we are sorry this contribution wasn't clearer before.
>
> - **Need clearer takeaways from theory (Q1)**.
>
> We apologize for the unclear presentation of our theory results. We added short 'Key takeaways' section at the end of Section 5 (theory section) summarizing the main theoretical consequences. We hope this provides greater clarity.
>
> - **Evaluation metrics beyond loss (Q2)**.
>
> We thank the reviewer for this suggestion. We agree that adding additional evaluation metrics would be helpful, but have struggled to develop reasonable evaluation procedures for model accuracy on these datasets during the revision period. We've considered LLM-as-a-judge and tried several types of prompt engineering and exact matching schemes, but are not fully confident in the accuracy of our approaches. We will note this as important future work.
>
> If the reviewer is interested, we have expanded our experimental evaluations by adding current results on K=3 for Llama 3.2 1B models and larger Llama 3.2 3B models for the $K=2$ settings. New experimental results can be found in Table 2 column (c) and Table 4. Table 2 (c) shows the results for interactions between k=2 larger llm models (Llama 3.2 3B) at $\beta=0.5$. Table 4. shows new results for extending the Llama 3.2 1B result from $K=2 \rightarrow K=3$ at $\beta=0.5$ and comparing it with our results for $K=3$ smaller OPT3 models. We also updated our experimental setup section to explain the task specific dataset for the third company being AI2 ARC dataset of science reasoning questions. Our additional results echo previous finding that cross-model training can both diversify and homogenize generative models as we scale up model size and including diversity by increasing $K$.

---

> > ### Comment · Reviewer_1aha · 2025-11-23
> >
> > Thanks for the updates! It seems the other reviewers had similar reactions to the paper. I think the clarifications and shortening mentioned will strengthen the paper. This still feels like a marginal accept to me given some of the limitations of the model and experiments that the other reviewers have highlighted.
> >
> > Regarding limited new dynamics in the multi-model setting, I just mean that the observed dynamics are what we would expect given the intuition from mode collapse in the single model setting. Homogenization + cross-model transfer is just like mode collapse around the "mean" of the multiple models: they all get more similar to each other, which has the effect of improving performance on each others' data distributions. It's nice to have theory and experiments to confirm this intuition (hence why I continue to think this paper is an "accept") but it's not a dramatic departure from the intuition derived from the single-model case. Of course, reality is not always surprising, and science is about discovering what is true rather than what is surprising. So it's still a nice contribution.

---

### Official Review · Reviewer_Z5bc · 2025-10-31

**Soundness:** 2
**Presentation:** 3
**Contribution:** 2
**Rating:** 4
**Confidence:** 4

**Summary:**

This paper investigates the phenomenon of data-mediated interactions among different generative AI models. Specifically, it studies what happens when different generative models are trained on each other’s outputs, a realistic scenario given the increasing prevalence of AI-generated content on the internet. The authors first review evidence that modern large language models (LLMs) are trained on overlapping, internet-scraped datasets that increasingly contain synthetic text from other models. Building on this, they formalize an iterative, interactive training framework where multiple entities train models using mixtures of public, private, and synthetically generated data. Then the authors give a theoretical analysis in a linear regression setting and derives closed-form dynamics for bias, variance, and convergence properties, showing that cross-model data sharing can promote homogenization while sometimes improving efficiency. Experiments using OPT-350M and LLaMA 3.2-1B fine-tuned on distinct tasks (SciQ and GSM8K) simulate multi-model interactions and confirm theoretical predictions: moderate mixing ($\alpha = \beta = 0.5$) yields the best balance.
The paper concludes that recursive cross-model training can both diversify and homogenize generative models.

**Strengths:**

- A novel problem. The paper's most significant contribution is the formalization of a new unstudied problem: "data-mediated interaction" within a multi-model systems. This shifts the research focus from the standard "model collapse" setting (a single model consuming its own outputs) to a more realistic and complex scenario where multiple heterogeneous models coexist and interact by training on a shared data pool containing each other's outputs.
- Comprehensive Methodology: The paper supports its claims with a comprehensive methodological approach that provides both theoretical and empirical evidence. The authors develop a formal theoretical model (a linear regression setting) to make analytical predictions about the system's long-term dynamics and (2) validate these predictions with a set of well-designed experiments using large language models (OPT-350m and Llama 3.2 1B).

**Weaknesses:**

- Drawback of the whole setting. The theoretical and empirical framework assumes that fine-tuning data for each model is randomly sampled according to ($\alpha-\beta$) proportions (new vs. old, public vs. private). However, in practice, major foundation models rely heavily on highly curated, high-quality fine-tuning datasets that are explicitly designed to avoid noise or low-quality synthetic data. This mismatch between the model’s random-mixing assumption and real-world fine-tuning practices limits the external validity of the results — particularly the conclusions about homogenization and performance degradation under synthetic data reuse.
- The theoretical analysis relies entirely on a linear regression model with Gaussian assumptions, which limits its generality to real-world large-scale nonlinear generative models. Although the authors cite “universality” results, the mapping from this toy model to practical LLM training remains speculative.
- After examining the released code (sft-config), it appears that each fine-tuning round uses a very small effective data volume: the batch size is 2, with gradient accumulation over two steps and only 100 optimization steps per generation. This means that each “generation” sees at most a few thousand training tokens, which is extremely small compared to realistic fine-tuning scales for modern LLMs. Consequently, the observed trends in “cross-model interaction” may reflect under-trained or noisy optimization dynamics rather than genuine long-term convergence effects. Moreover, the experiments involve only K = 2 interacting models and explore just three discrete values for both $\alpha$ and $\beta$ ({0, 0.5, 1}), providing too coarse a sampling to fully characterize the theoretical phase behavior. These limitations substantially weaken the empirical support for the paper’s broader claims about multi-model ecosystems.
- The paragraph leading with Step 3: Model Updates (lineno 180-188)' is misleading. The paper’s description of “model updates” incorrectly claims that successive generations of models such as GPT-1/2/3/4 and LLaMA-1/2/3 are typically trained by initializing with the previous generation’s weights. Only using the same datasets to train a family of models does not imply directly descend from one another'.
- The proofs contain several typographical and notational issues that impede verification and, in a few places, likely invalidate steps. See more in Questions.

**Questions:**

- Please kindly think of Weakness 1. The interaction of different models is a novel problem, while the analysis framework in this paper is a little weak. Could you improve the problem setting and make it closer to the reality?

- There is a grammar mistake in the paper's title. Use \textbf{each other's} instead of \textit{each others'}.
- I have a question on the proof of Lemma 1 (Appendix E.4). In line 1020, the derivation implicitly equates $(\Pi S \Pi)$ with $(I_K \otimes \underline{S})$. These two matrices are not equal. $(\Pi S \Pi)$ is a dense block matrix (specifically $\frac{1}{K}(J \otimes \underline{S})$). $(I_K \otimes \underline{S})$ is a block-diagonal matrix. If the equation holds for the spectrum norm, I'd appreciate it if you can provide a detailed calculation.

- Typos. Here I list several obvious mistakes and please proofread the whole paper to improve the quality of presentation.

- Line 276: its most recent parameter estimate $\hat\theta_{t−1,k}$ instead of $\hat\theta_{t−1,t}$.
- Line 291. In the rightmost, $y_{t1}$ instaed of $y_{tk}$.
- Different definition of $S_*$ in the main text Section 5 and appendix E.

---

> ### Author Response · Authors · 2025-11-22
>
> We thank the reviewer for their helpful feedback and respond to their points below. We have revised our paper to address the reviewer's concerns and will point them to appropriate revisions throughout our response. Revised paper text is teal.
>
> - **Grammar issues**
>
> Thanks for catching! We've updated our title.
>
> - **Limitations of Model**. Our framework is not limited to random sub-sampling and it  can be used study the impact of high-quality private data.  To simplify the presentation we focus on the key tradeoffs  described by the parameters $\alpha$ and $\beta$, but many other aspect can be studied directly under our linear dynamical setup.
>
>   - **Sub-sampling**: Our framework allows random sub-sampling but does not require it. Note that in Section 5, the estimate at each stage is based on weighted average of all previous data. In particular, we may ensure that all the original data is used at every step.
>
>   - **Different noise level**: The impact of different types of data (high noise vs. low noise) can be captured naturally by rescaling the feature matrices and adjusting the weights accordingly. The result is that the high-quality data has a much larger influence on the training dynamics, as described by our main theorems.
>
> - **Linear regression**. The focus on linear regression is a limitation of our theory with respect to the target applications in LLMs (and this is why empirical results are so important). Nevertheless, we firmly believe that that such analysis meaningful:
>
>   - Our setup builds upon and extends a large body of existing and ongoing theoretical work that focuses directly on linear regression settings. The recent work of Dey and Donoho does a good job of describing the universality class to which the theory applies directly --- namely, high-dimensional generalized linear models satisfying classical asymptotic normality conditions. This class is broader than just linear regression but it still falls short of modern machine learning models. Their use of these models is a key motivation for our own use of them.
>   - Our work demonstrates that a linear dynamical system model is able to capture a wide range of behaviors (both positive and negative) that can arise within the interactive retraining paradigm. Moreover, we suspect that any attempt to theoretically analyze learning in a setting that does not allow for such dynamics would be inherently limited.
>
> - **Small effective data volume in experiments**. Thanks for actually looking at our code! Yes, the effective batch size in our fine-tuning experiments is 4. Batch size was set this low due to OOM issues with our GPUs when working with these very large models. We've since found that we can increase the effective batch size to 16 without encountering OOM issues. We've run the $K=3$ experiments in the revision with this higher batch size, and we plan to update the $K=2$ experiments to use this batch size as well. We simply lack the time to re-run $K=2$ and also produce $K=3$ results during the revision period. Also, we calculate that the models see roughly 19K tokens per fine-tuning step, which is small but, we believe, not unreasonable.
>
>   New experimental results can be found in Table 2 column (c) and Table 4. Table 2 (c) shows the results for interactions between k=2 larger llm models (Llama 3.2 3B) at $\beta=0.5$. Table 4. shows new results for extending the Llama 3.2 1B result from $K=2 \rightarrow K=3$ at $\beta=0.5$ and comparing it with our results for $K=3$ smaller OPT3 models. We also updated our experimental setup section to explain the task specific dataset for the third company being AI2 ARC dataset of science reasoning questions. Our additional results echo previous finding that cross-model training can both diversify and homogenize generative models even when we scale up model size and diversity by increasing $K$.
>
> - **Misleading wording about model updates**. We thank the reviewer for catching this imprecise wording. We have removed this sentence from Section 3 implying direct weight inheritance across model generations (not visible in revision since we removed text rather than added it).

---

> ### Author Response · Authors · 2025-11-22
>
> - **Proof issues and typos**. Thanks for catching the typos and the carefully reading of the proof of Lemma~1. We agree that a few more lines of explanation would be helpful here. Please note that that the equality (in line 1020 of original submission) is with respect to the spectral norms of the matrices. Since two matrices can be very different and yet have the same spectral norm, this equality neither implies nor requires equality at the matrix level.
>
>     Importantly, the matrices $\Pi S \Pi$ and and $I_K \otimes \underline{S}$ have the same non-zero eigenvales, which by symmetry are the same as the singular values. To see why, observe that
>
>      $$ \Pi S \Pi =  \tfrac{1}{K} \boldsymbol{1}_{K \times K}  \otimes \underline{S} $$
>
>     The matrix $\tfrac{1}{K} \boldsymbol{1}_{K \times K} $ is a projection matrix so all of its eigenvalues are 0 or 1. Since the eigenvalues of the Kronneker product of two matrices are the pairwise products of the eigenvalues of the individual matrices, it follows that  $\Pi S \Pi$ and $I_K \otimes \underline{S}$ have the same nonzero eigenvalues values.
>
>     In the revision, we have updated the derivation as follows:
>     $$  || \Pi R \Pi || =^{(a)}  \| \Pi ( I_K + \Delta^{+/2}) S  ( I_K \otimes  \Delta^{+/2}) \Pi\|$$
>     $$  \hspace{1.6cm} =^{(b)}  \| ( I_K \otimes  \Delta^{+/2}) \Pi S \Pi ( I_K \otimes  \Delta^{+/2})\|$$
>     $$  \hspace{1.6cm} =^{(c)}  \| ( I_K \otimes  \Delta^{+/2}) (\tfrac{1}{K} \boldsymbol{1}_{K \times K}  \otimes \underline{S}  )  ( I_K \otimes  \Delta^{+/2})\| $$
>
>      $$  \hspace{1.6cm} =^{(d)} \| \tfrac{1}{K} \boldsymbol{1}_{K \times K}  \otimes \Delta^{+/2} \underline{S} \Delta^{+/2}\|$$
>
>      $$  \hspace{1.6cm} =^{(e)} \| \tfrac{1}{K} \boldsymbol{1}_{K \times K} \| \|  \Delta^{+/2} \underline{S} \Delta^{+/2}  \| $$
>
>      $$  \hspace{1.6cm} =^{(f)}  \|  \Delta^{+/2} \underline{S} \Delta^{+/2}  \| $$
>
> where (a) is the definition of $R$; (b) follows from the commutativity of $\Pi$ and $I_K \otimes \Delta^{+/2}$; (c) follows from  $\Pi S \Pi = \frac{1}{K} \boldsymbol{1}_{K \times K} \otimes \underline{S}$;
>
> (d) is the mixed-product property of the Kronneker product; (e) is the basic identity $\|A \otimes B\| = \|A\| \|B\|$ for any matrices $A$ and $B$; and (f) holds because $\| \tfrac{1}{K} \boldsymbol{1}_{K \times K} \| = 1$. Finally, using that $0 \preceq \underline{S}\preceq \Delta$, we see that $ \|  \Delta^{+/2} \underline{S} \Delta^{+/2}  \|\le  \|  \Delta^{+/2} \Delta  \Delta^{+/2}  \| \le 1$.

---

### Official Review · Reviewer_rQht · 2025-11-01

**Soundness:** 2
**Presentation:** 4
**Contribution:** 2
**Rating:** 4
**Confidence:** 2

**Summary:**

The paper studies how generative language models (LLMs) may interact recursively through training data that include other models’ outputs.

1. It introduces a formal framework with two parameters — the synthetic data ratio ($\alpha$) and the initial data weight ($\beta$) — to describe cross-model data mixing.

2. Theoretical analysis (Sec. 3) under a generalized linear model shows convergence and bias–variance behavior depending on $\alpha$ and $\beta$.

3. Empirical results (Sec. 4; Fig. 3–5) using OPT-350M and LLaMA-1B demonstrate that moderate mixing ($\alpha=\beta=0.5$) improves both models’ performance but leads to representation homogenization.

The paper also concludes with a discussion of the implications for model diversity and long-term ecosystem dynamics (Sec. 5).

**Strengths:**

1. Clear and meaningful problem setting：The paper addresses a timely and practically significant question — how recursive data interactions among generative models affect learning stability and diversity. The motivation and background are well-articulated (Sec. 1–2), making the research goal both relevant and understandable.

2. Comprehensive and interpretable theoretical framework：The proposed formalism based on the parameters $\alpha$ and $\beta$ (Sec. 3) systematically captures cross-model data mixing. The accompanying bias–variance and convergence analysis provides solid conceptual grounding for the empirical findings (Fig. 2–3). Even without verifying every derivation, the overall reasoning is coherent and accessible.

3. Exceptional clarity and readability：The writing is well-structured and accessible to readers beyond the immediate subfield. Explanations, figures, and notation are consistently clear, enabling a broad audience to grasp the motivation, methodology, and conclusions (Sec. 1–5).

**Weaknesses:**

1. Limited novelty in the modeling of cross-model interaction：The description of data-mediated interactions between models (Sec. 3) is clear and well-structured but largely descriptive. While it helps readers understand the setup, this section mainly formalizes an intuitive process rather than introducing a new mechanism or theoretical insight. As a result, the contribution of this part feels limited in terms of originality.

2. Gap between theoretical modeling and practical relevance：Most of the paper focuses on theoretical modeling and proofs (Sec. 4–5). Although the derivations appear sound, the connection to real-world large-scale training scenarios remains weak. The introduction of parameters $\alpha$ and $\beta$ is conceptually useful, yet in practice, their exact values or ratios are difficult to estimate or control during continuous training. The conclusions drawn from the linear or generalized linear setting may not easily transfer to nonlinear or high-dimensional models.In essence, while the problem definition is good and $\alpha$–$\beta$ reasoning is meaningful, it is unclear how the theory can concretely guide actual large-model training.

3. Experimental validation is narrow and idealized：The experiments (Sec. 4–5) mainly serve to verify the theory, but they do not provide further insights into realistic settings. Only two medium-sized models (OPT-350M and LLaMA-1B) and two datasets (SciQ, GSM8K) are used, with highly controlled data composition. The synthetic data are assumed to represent model outputs cleanly, without considering realistic mixtures of human and synthetic text (I know in limitation part). Scaling experiments or additional ablations (e.g., varying model size, task diversity, or realistic data proportions) would make the findings more convincing.

Overall, the experimental content is rather insufficient. The question itself is meaningful, but it does not provide much insight in terms of conclusions. However, considering that this might be a theoretical paper, it is difficult to for me to  assess the practical value of such a theory. Therefore, I would lower the confidence to mitigate the possible impact of this uncertainty.

**Questions:**

You may refe to the content in the “weakness” section. If you can address the doubts raised there effectively, I will consider giving a higher score.

I hope valuable work will not be overlooked.


For example, in addition to theoretical explanations based on existing assumptions, it would be great if you could highlight some unique insights proposed in this paper.
Perhaps the paper already includes them, but I did not notice.

---

> ### Author Response · Authors · 2025-11-22
>
> We thank the reviewer for their helpful feedback and respond to their points below. We have revised our paper to address the reviewer's concerns and will point them to appropriate revisions throughout our response. Revised paper text is teal.
>
> - **Limited novelty in modeling (Weakness 1).**
>
> We appreciate the reviewer's point but respectfully disagree. First, our model covers new ground in the model collapse space: introducing data-mediated interactions between models in a recursive training setting. As Section 3 of our paper firmly establishes, this setup has not been addressed in prior work. Beyond this, we introduce a two-parameter formal framework ($\alpha,\beta$) that cleanly decomposes the effects of new vs. reused data and public vs. private data. This enables analytical study of model behaviors under interactive, recursive training, which has not been previously possible (nor even proposed for study). Finally, our practical LLM experiments provide the first empirical confirmation of these dynamics in practice. In the revision we have added a short paragraph highlighting these novel aspects (see "Key Insights" at the end of the introduction).
>
> - **Gap between theory and large-scale practice (Weakness 2).**
>
> We agree that linear/Gaussian analysis is idealized, but it offers a mathematically transparent setting to isolate and explain the core dynamics observed empirically. All prior literature studying model collapse has used similar simplifying assumptions. We have expanded a discussion of these limitations in the revised version of the paper (see Discussion section), added citations to prior literature leveraging such simplified models (see start of Theory section) and identifying potential disconnects between our theory and practical LLM behaviors (again, in Discussion).
>
> - **Narrow experiments.**
>
> We will do our best to expand the experimental discussion. We do have new results for $K=2$ on a larger Llama 3.2 (3B) models and $K=3$ Llama 3.2 (1B) models that we have added in the paper revision (for $T=8$). Furthermore, we have added a table of fine-tuning hyperparameters and seeds (in Appendix) and clarify that the current experiments use small-scale fine-tuning as controlled, interpretable probes rather than attempting to mimic industry-scale re-training.  Thanks for the suggestions about ablating over the fraction of synthetic vs. real data and add noise-robustness checks where generated outputs are mixed with real human-authored text. We will note these in the paper as interesting future work, as we lack sufficient time to complete these experiments during the rebuttal.
>
> New experimental results can be found in Table 2 column (c) and Table 4. Table 2 (c) shows the results for interactions between k=2 larger llm models (Llama 3.2 3B) at $\beta=0.5$. Table 4. shows new results for extending the Llama 3.2 1B result from $K=2 \rightarrow K=3$ at $\beta=0.5$ and comparing it with our results for $K=3$ smaller OPT3 models. We also updated our experimental setup section to explain the task specific dataset for the third company being AI2 ARC dataset of science reasoning questions. Our additional results echo previous finding that cross-model training can both diversify and homogenize generative models even when we scale up model size and diversity by increasing $K$.
>
> - **Request to highlight unique insights.**
>
> Thank you for pointing out this oversight. We have added a "key insights" paragraph at the end of the introduction to better emphasize this. To briefly summarize here, we find, both theoretically and empirically, that moderate cross-model mixing can improve model performance on otherwise unseen-data but risks homogenizing model behavior.

---

### Author Response · Authors · 2025-12-03
**Comments for the Area Chair**

Dear Area Chair,

We thank you and all reviewers for spending time evaluating our submission. We sincerely appreciate the constructive feedback, and we are grateful that several reviewers found the paper well-executed, timely, and deserving of publication, as well as a meaningful contribution to the area of model collapse.

- **Main updates made in the revision:** (1) We clarified the novelty and contribution of our work by adding a "key insight" section in the introduction that distinguishes our multi-model framework from prior single-model collapse studies. (2) We expanded the theory discussion on our assumptions and limitations of linear models and emphasized that our work demonstrates a linear dynamical system is able to capture a wide range of behaviors. (3) We added new results with larger models (Llama 3.2-3B), more models (K=3) interacting with diverse tasks, and larger effective batch sizes. We also added hyperparameter tables and clarified compute tradeoffs and future plans for larger-scale runs. (4) Corrected typos, proofs, and updated our references to be more formal with peer-reviewed sources.

Reviewer rQht stated that they would consider raising their score if their concerns were adequately addressed. Accordingly, we expanded our novelty discussion, clarified the practical relevance of our theory, added new experimental evidence by scaling up model size and number of models interacting, and included explicit key insights. These updates directly target the reviewer’s stated conditions for improving their evaluation, and we believe they fully satisfy the concerns raised.

---

### Meta-Review · Area_Chair_jneC · 2026-01-05

**Summary:**

The paper investigates the phenomenon of "data-mediated model interactions" in a multi-model ecosystem. In particular, the paper studies the scenario where multiple distinct models are trained on a shared pool of data from the internet that increasingly contains data generated by other models. The authors present a theoretical framework with two key parameters that capture the ratio of private vs public data in the initial dataset, and the ratio of model-generated data to human data in each update. They then develop a theoretical analysis using a generalized linear model to predict bias, variance and convergence. They also conduct an empirical investigation (that has been expanded upon during the rebuttal phase) that is intended to simulate recursive training over multiple generations of models.


The main concerns that the reviewers brought up are:
- **C1**. Limited novelty in the modeling of cross-model interaction. The mechanism proposed is not fundamentally different from previous ones that consider a single model, and does not yield a fundamentally new theoretical insight (Reviewer rQht).
- **C2**. Gap between theoretical modeling and practical relevance. The conclusions from analysis in generalized linear model may not transfer to practical scenarios; unclear how to compute alpha and beta in practice (Reviewer rQht, Reviewer Z5bc).
- **C3**. Experimental validation is limited in scope and overly simplistic. Both Reviewer rQht and xD3M point out that only two medium-sized models are used, and only two datasets, with highly controlled data composition. Reviewer xD3M also points out that T=15 is too small (should consider values up to 100 iterations) and that the experimental setup lacks realism in terms of not accounting for training data quality and task diversity.
- **C4**. Unrealistic assumption of randomly sampling data to finetune on, as opposed to finetuning on highly-curated high-quality data, as is done in practice (Reviewer Z5bc).
- **C5**. Implementation issue of small batch size, deviating from conditions in which realistic fine-tuning occurs in modern models (Reviewer Z5bc).
- **C6**. Limited insights compared to what we expected based on prior work in the single-model setting (Reviewer 1aha).
- **C7**. Writing improvements: the manuscript can be improved by compiling the main key take-aways more explicitly, and presenting the main results earlier in the paper (Reviewer 1aha, Reviewer xD3M).

**Reviewer Concerns:**

Some of the above concerns remain unaddressed, such as (i) the inherent limitation of the simplistic theoretical framework to capture realistic LLM models, and (ii) the weakness of the empirical investigation, in terms of the lack of scalability (K and T values too small to offer insights into practical settings), too coarse-grained settings (not enough values of alpha and beta),  the implementation issue of having used an overly small batch size (due to OOM issues the authors faced but have since resolved), and failing to capture practically important factors like task diversity and data quality.

Some of the above weaknesses are acceptable, e.g. the authors make a reasonable argument that their theoretical framework extends the same idealized setting that is commonly used in the literature and is still meaningful for analyses and interpretability, despite simplifications. It also may be acceptable not to reach K=50 and T=100 (though the experimental section could have still been reasonably strengthened had the authors expanded the scope of their experiments).

Overall, despite the remaining limitations of the empirical evaluation, the paper has several strong points and makes a good contribution. It tackles a timely and novel research problem, providing both theoretical and empirical investigations, and having exceptional clarity and readability. Therefore, I believe the paper can have a positive impact on the community, and I recommend acceptance. I strongly encourage the authors to incorporate any experiments they have obtained in the meantime (they mentioned they are running or planning to run additional experiments that may not complete before the rebuttal phase) into the updated paper.

**Reviewer Scores:**

**Reviewer rQht**.
To address C1, the authors argue that their theoretical framework that decomposes the effect of new vs reused data and public vs private data enables an analytical study of model behaviours within the multi-model ecosystem, which was not previously possible. I find this to be a compelling argument and the reviewer may have agreed, especially since this reviewer acknowledged that the setup studied here is meaningful, timely and well-motivated. In my view, even if the mechanism proposed isn’t novel, there is novelty from the investigation of a previously-uninvestigated (and meaningful) research question.

For C2, the authors acknowledge that the analysis is under idealized conditions but argue that it is still meaningful to isolate and explain core dynamics. Further, it is common in this literature to make this simplifying assumption for theoretical analysis. This seems reasonable to me given that this is standard practice and lifting these simplifications leaves us with an intractable setting.

To address C3, the authors added new experiments for K=2 on a larger (3B) LLama model and added new experiments for K=3 for 1B models, showing that these new results are in line with previous findings.
Other ablations that the reviewer suggested are left for future work.
In light of this, the reviewer may have maintained the position that, overall, the experimental content of the paper is weak and we have not gained much insight from it, aside from the unsurprising conclusion that moderate mixing leads to some positive transfer but can also cause some homogenization.


**Reviewer Z5bc**.
To address C4, the authors claim that their framework is not limited to the random subsampling procedure but allows the use of weighing of different data. While this is reassuring, my understanding is that the “more realistic” setting that the reviewer describes of prioritizing high-quality data for finetuning has not been investigated empirically, making the experimental setup a poor proxy for real-world scenarios from this perspective too.

For C2, the authors reiterate that their linear setup does have limitations but remains meaningful to analyze and it extends a large body of theoretical work focusing on this setting. This seems reasonable to me.

For C5, the authors admit that they used a very small batch size due to OOM issues (which they have since fixed but did not get the chance to rerun the old K=2 experiments since the fix was applied). The fact that the original results were run in this suboptimal (unrealistic) setting is an outstanding weakness of the empirical investigation.

In light of these unresolved issues, it seems possible that the reviewer would have maintained their score of a weak reject.


**Reviewer 1aha**.

For C6, the authors argue that their setting is distinct from the single-model setting due to the possibility of positive transfer that exists here, between private data of one model towards different models (that indirectly benefit through other models’ private data due to training on synthetic data generated by those other models). The reviewer agreed with this but clarified that this can perhaps be seen as a model collapse towards the “mean” of different models in this case, not too different from the single model case. However, the reviewer concedes that this was not known ahead of time, so generating this knowledge is valuable even if the result isn’t surprising.
The authors have addressed C7 well in the revised manuscript.
In response to the reviewer’s question about accuracy (rather than just loss), the authors responded that they did not have time to explore additional metrics.
Overall, the reviewer is positive towards the submission, despite the limitations of the theoretical analysis and shortcomings of the empirical evaluation, as the reviewer feels that these findings are interesting and valuable.


**Reviewer xD3M**.
The authors addressed some of the reviewer’s comments well (e.g. C7 is addressed in the revised manuscript).
The main outstanding concern is the narrow scope of the empirical evaluation. While the reviewer wants K>10 and T>50, the authors initially presented K=2 and T=15, and in the rebuttal expanded to K=3 but only with T=8.
The authors did not address the lack of realism of their experimental setting due to not accounting for data quality or task diversity.
The reviewer weakly recommends acceptance.

---

### Decision · Program_Chairs · 2026-01-26

Accept (Poster)